

# The importance of freshwater systems to the net exchange of atmospheric carbon dioxide and methane with rapidly changing high Arctic landscapes

Craig A. Emmerton[1], Vincent L. St. Louis[1], Igor Lehnherr[2], Jennifer A. Graydon[1], Jane L. Kirk[3], Kimberly J. Rondeau[1]

[1]Department of Biological Sciences, University of Alberta, Edmonton, Alberta T6G 2E9, Canada
[2]Department Geography, University of Toronto, Mississauga, Ontario, L5L 1C6, Canada
[3]Science and Technology Branch, Environment Canada, Burlington, Ontario, L7R 4A6, Canada

*Correspondence to*: Craig A. Emmerton (emmerton@ualberta.ca)

**Abstract.** A warming climate is rapidly changing the distribution and exchanges of carbon within high Arctic ecosystems. Few data exist, however, which quantify exchange of both carbon dioxide ($CO_2$) and methane ($CH_4$) between the atmosphere and freshwater systems, or estimate freshwater contributions to total catchment exchange of these gases, in the high Arctic. During the summers of 2005 and 2007-2012, we quantified $CO_2$ and $CH_4$ concentrations in, and atmospheric exchange with, common freshwater systems in the high Arctic watershed of Lake Hazen, Nunavut, Canada. We identified four types of biogeochemically-distinct freshwater systems in the watershed, however mean $CO_2$ concentrations (21–28 µmol L$^{-1}$) and atmospheric exchange (-0.013–0.046 g C-$CO_2$ m$^{-2}$ d$^{-1}$) were similar between these systems. Seasonal flooding of ponds bordering Lake Hazen generated considerable $CH_4$ emissions to the atmosphere (0.008 g C-$CH_4$ m$^{-2}$ d$^{-1}$), while all other freshwater systems were minimal emitters of this gas (<0.001 g C-$CH_4$ m$^{-2}$ d$^{-1}$). Measurements made on terrestrial landscapes in the same watershed between 2008-2012 determined that the near-barren polar semidesert was a very weak consumer of atmospheric $CO_2$ (-0.004 g C-$CO_2$ m$^{-2}$ d$^{-1}$), but an important consumer of atmospheric $CH_4$ (-0.001 g C-$CH_4$ m$^{-2}$ d$^{-1}$). Alternatively, meadow wetlands were very productive consumers of atmospheric $CO_2$ (-0.96 g C-$CO_2$ m$^{-2}$ d$^{-1}$) but relatively weak emitters of $CH_4$ to the atmosphere (0.001 g C-$CH_4$ m$^{-2}$ d$^{-1}$). When using ecosystem-cover classification mapping, we found that freshwaters were unimportant contributors to total watershed carbon exchange, in part because they covered less than 10% of total cover in the watershed. High Arctic watersheds are experiencing warmer and wetter climates than in the past, which may have implications for the net uptake of carbon greenhouse gases by currently underproductive polar semidesert and freshwater systems.

*Keywords:* carbon dioxide, methane, lake, pond, high Arctic, climate change, watershed





## 1 Introduction


Freshwater ecosystems cover less than 10% of global ice-free land area (Lehner and Doll, 2004) and have
been typically overlooked as substantial contributors to, or sinks of, atmospheric carbon greenhouse gases (GHGs;
Bastviken et al., 2011). However, recent studies suggest inland lakes collectively receive and process carbon at
magnitudes similar to oceanic uptake and sediment burial, making them important systems within the global carbon
cycle (Cole et al., 2007; Battin et al., 2009; Tranvik et al., 2009; Maberly et al., 2013; Raymond et al., 2013).
Though these lowland systems efficiently accumulate allochthonous and autochthonous carbon, most natural lakes
and ponds, on balance, emit carbon GHGs to the atmosphere. For example, in most freshwater ecosystems,
decomposition continuously adds carbon dioxide ($CO_2$) to the water before venting to the atmosphere (Rautio et al.,
2011), while uptake of $CO_2$ by autotrophs occurs typically over shorter seasonal periods (Cole et al., 2000; Huttunen
et al., 2003; Breton et al., 2009; Bastviken et al., 2011; Rautio et al., 2011; Callaghan et al., 2012). At the same time,
lake sediments and even oxic waters can sustain bacterial methanogenesis and the production of the potent GHG
methane ($CH_4$; Bogard et al., 2014). Due to the gas's poor solubility in water, ebullition and wind can then
efficiently vent $CH_4$ to the atmosphere from these ecosystems, perhaps contributing up to 12% of global emissions
(Lai, 2009; Walter et al., 2006).
Lakes, ponds and wetlands are globally most abundant in northern regions, largely due to past periods of
glaciation and resulting land deformation. These freshwater environments may cover greater than half the landscape
in northern regions, and can account for more than three-quarters of a landscape's net $CO_2$ exchange with the
atmosphere (Abnizova et al., 2012). Saturated northern peatlands can also be robust emitters of $CH_4$ because
permafrost impedes drainage of soils, promoting anoxia and methanogensis (Tagesson et al., 2012; Wik et al.,
2016). However, at the highest northern latitudes (>70°N), polar semidesert landscapes not covered by glaciers
generally have cold, relatively well-drained soils (Campbell and Claridge, 1992) and receive little precipitation,
resulting in often less than 5% of the landscape being covered by aquatic systems. Though easy to overlook as
sparsely-vegetated barrens that exchange few carbon GHGs with the atmosphere (Soegaard et al., 2000; Lloyd,
2001; Lund et al., 2012, Lafleur et al., 2012), recent studies have shown that where conditions are ideal, high Arctic
ecosystems exchange GHGs at rates similar to ecosystems at more southerly latitudes (Emmerton et al., 2014;
Emmerton et al., 2016).



56   High Arctic ecosystem productivity is currently changing rapidly as a warming climate substantially alters

57 polar watersheds (IPCC, 2007a). Some climate models predict that in the Canadian Arctic, autumn and winter

58 temperatures may rise 3-5$^{o}$C by 2100, and up to 9$^{o}$C in the high Arctic (>70ºN; ACIA, 2004; IPCC, 2007b). Mean

59 annual precipitation is projected to increase ~12% for the Arctic as a whole over the same period, and up to 35% in

60 localized regions where the most warming will occur (ACIA, 2004; IPCC, 2007b). Such warming and wetting is

61 already modifying Arctic landscape energy balances (Froese et al., 2008) resulting in glacial melt (Pfeffer et al.,

62 2008), permafrost thaw (Peterson et al., 2002), reorganized hydrological regimes (i.e., drying or wetting; Smith et

63 al., 2008) and extended growing seasons (Manabe et al., 1994). These changes are also perturbing watershed carbon

64 cycling through, for example, the liberation of carbon from thawing permafrost, and increases in biological

65 productivity on landscapes and in lakes, ponds and wetlands (Mack et al., 2004; Smol et al., 2005; Walker et al.,

66 2006; Smol and Douglas, 2007). Considering the extensive cover of the near-barren polar semidesert in the high

67 Arctic (>10$^6$ km$^2$), these changes may have considerable effects on the future net exchange of carbon GHGs both

68 locally and on a global scale (Anthony et al., 2014). However, climate changes are far better delineated and

69 predicted for high Arctic landscapes in general than specifically for freshwater systems and landscape-scale GHG

70 exchange with the atmosphere. Therefore it is uncertain how rapid climate change will alter the cycling of carbon in

71 these remote regions.

72   The primary objective of this study was to measure the net atmospheric exchange of $CO_2$ and $CH_4$ with

73 common high Arctic freshwater ecosystems, and place these findings in context with recent studies of terrestrial

74 GHG exchange from this remote polar semidesert biome.

75 **2 Methods**

76 **2.1 Location and sampling overview**

77   We conducted our research at the Lake Hazen base camp in central Quttinirpaaq National Park, Ellesmere

78 Island, Nunavut (81.8º N, 71.4º W), Canada's most northerly protected area (Figure 1). Lake Hazen (area: 542 km$^2$;

79 max. depth: 267 m) is the world's largest high Arctic lake, and is surrounded by a substantial watershed (6,901 km$^2$).

80 About 42% of the Lake Hazen watershed is glaciated with the balance of area covered by a polar semidesert (>90%

81 of ice–free area; Edlund, 1994), small lakes, ponds and meadow wetlands. The lower Lake Hazen watershed is a

82 high Arctic thermal oasis (France, 1993) as it experiences anomalously warm growing season (June–August; 92





days) conditions because it is protected from cold coastal weather by the Grant Land Mountains and Hazen Plateau
(Table S1). For example, mean July air temperature is typically 8-9 ℃ at the base camp, compared to July 1981-
2010 climate normals of 6.1 ℃ and 3.4 ℃ at the coastal Eureka and Alert weather stations on Ellesmere Island,
respectively (Environment Canada, 2016). Soils in the region are also atypically warm during the summer because
of low moisture content and efficient radiative heating due to an abundance of clear-sky days. These conditions,
coupled with continuous daylight during the growing season, have resulted in a greater diversity and abundance of
vegetation and wildlife in the Lake Hazen watershed compared to surrounding areas (France, 1993), despite
receiving only ~34 mm of precipitation during the growing season (Table S1). Ultra-oligotrophic Lake Hazen itself
dominates the freshwater area of the watershed (Keatley et al., 2007) and receives most of its water annually from
rivers discharging melt water from glaciers. Water exits Lake Hazen via the Ruggles River. Ice-cover can remain on
Lake Hazen throughout the growing season, though in recent years the lake has gone ice-free more frequently,
usually by late July. Ponds and a few small lakes are scattered throughout the lower watershed and are mostly
shallow, small in area (~70% are <1 ha) and typically go ice-free by mid- to late-June each year.
To quantify net GHG exchange of typical high Arctic freshwater bodies, we identified several ponds or
small lakes to sample within walking distance of base camp in the northwestern portion of the Lake Hazen
watershed (Figure 1). These systems were chosen systematically to incorporate a gradient of watershed position,
surface area, mean depth, emergent vegetation productivity, and hydrological connectivity (Table 1). We also
sampled just offshore in Lake Hazen to obtain water representative of that which interacted with ponds located along
its shoreline. All sampling occurred during the summer growing seasons of 2005 to 2012 (except for 2006), between
mid-June and early August (Table S2).
**2.2 Dissolved $CO_2$ and $CH_4$ concentrations of high Arctic freshwaters**
Two approaches were used to quantify concentrations of dissolved $CO_2$ and $CH_4$ in surface waters. The
first approach was employed at all sites and used the common method of collecting water directly into evacuated
160-mL Wheaton glass serum bottles capped with butyl rubber stoppers (Hamilton et al., 1994; Kelly et al., 1997).
Each bottle contained 8.9 g of potassium chloride (KCl) preservative to kill all microbial communities (Kelly et al.,
2001), and 10 mL of ultra high purity dinitrogen ($N_2$) as a gas headspace. To collect a sample, a bottle was
submersed ~5 cm below the water surface and punctured with an 18-gauge needle. Barometric pressure and water
temperature were recorded. Dissolved gas samples were stored in the dark at ~5°C until return to the University of





Alberta, where they were analyzed in the accredited Biogeochemical Analytical Service Laboratory (BASL). There,
samples were placed in a wrist-action shaker for 20 minutes to equilibrate dissolved $CO_2$ and $CH_4$ with the $N_2$
headspace. Headspace $CO_2$ and $CH_4$ concentrations were quantified on a Varian 3800 gas chromatograph (GC)
using a flame ionization detector at 250°C with ultra high purity hydrogen ($H_2$) as a carrier gas passing through a
hayesep D column at 80°C. A ruthenium methanizer converted $CO_2$ to $CH_4$. Four gas standards (Praxair, Linde-
Union Carbide), ranging from 75 to 6000 ppm for both $CO_2$ and $CH_4$, were used to calibrate the GC. A Varian Star
Workstation program integrated peak areas and only calibration curves with an $r^2$ >0.99 were accepted for analyses.
A standard was re-analyzed every 10 samples to reconfirm the calibration, and duplicate injections were performed
on all samples. Headspace $CO_2$ and $CH_4$ concentrations were converted to dissolved molar concentrations using
Henry's Law, and corrected for temperature and barometric pressure differences between sample collection and
analysis. To quantify dissolved inorganic carbon (DIC) concentrations, samples were acidified with 0.5 mL $H_3PO_4$
to convert all DIC to $CO_2$, and then immediately reanalyzed on the GC. DIC concentrations were calculated as
above.

The second approach involved two automated systems to determine detailed diel changes in surface water

dissolved $CO_2$ concentrations at two different sites (Skeleton Lake and Pond 01; Figure 1; Table S2). Dissolved $CO_2$
concentrations were measured every three hours during several summers. These systems functioned by equilibrating,
over a 20-minute period, dissolved $CO_2$ from pumped surface waters, with a gas cell in a Celgard MiniModule
Liqui-Cel. The equilibrated gas was then analysed for $CO_2$ concentration by a LI-COR (Lincoln, NE) 820 infrared
gas analyzer. The systems also measured dissolved oxygen ($O_2$) concentrations using a Qubit$^{TM}$ flow-through
sensor. Concentrations were then converted to aqueous molar concentrations using Henry's Law and water
temperature quantified with a Campbell Scientific (Logan, UT) 107-L thermistor. The systems were housed in
watertight cases along the shore from which a sample line extended out into the surface waters, and upon which was
mounted a CS 014A anemometer (1 m height) and a Kipp & Zonen (Delft, The Netherlands) photosynthetically-
active radiation (PAR) LITE quantum sensor. All data were recorded on Campbell Scientific CR10X dataloggers.
**2.3 Dissolved $CO_2$ and $CH_4$ concentrations of high Arctic freshwaters**

Though several models exist for quantifying turbulent gas fluxes of lakes (e.g., MacIntyre et al., 2010), we

decided to use the stagnant film model described by Liss and Slater (1974) to quantify net $CO_2$ and $CH_4$ mass fluxes
between surface waters and the atmosphere at our remote location. This decision was made because 24-hour



daylight at our high-latitude location dampened diurnal surface temperature changes to less than 1°C, the general
shallowness of the systems, and the steady, sometimes gusty, wind conditions on site. The stagnant film model
assumes gas concentrations in both surface waters and the atmosphere are well-mixed, and that gas transfer between
the phases occurs via diffusion across a diminutive stagnant boundary layer. Diffusive gas transfer across the
boundary layer is assumed to follow Fick's First Law:
$$\text{Gas flux } (\mu mol\ m^{-2}\ hr^{-1}) = k(C_{SUR} - C_{EQL}) \tag{1}$$

where $C_{SUR}$ ($\mu mol\ L^{-1}$) is the concentration of the gas in surface waters, $C_{EQL}$ ($\mu mol\ L^{-1}$) is the atmospheric
equilibrium concentration, and k is the gas exchange coefficient, or the depth of water per unit time in which the
concentration of the gas equalizes with the atmosphere (i.e., piston velocity). Values of k ($cm\ hr^{-1}$) were calculated
using automated systems wind measurements and occasionally from nearby (within 2 km) eddy covariance towers
(Campbell Scientific CSAT3 Sonic Anemometers; 30 min. means), and published empirical relationships (Table S3;
Hamilton et al., 1994). To determine the direction of the flux, atmospheric equilibrium $CO_2$ and $CH_4$ concentrations
were quantified using Henry's law, in-situ barometric pressure and air temperature, and mean annual $CO_2$ and $CH_4$
concentrations in the atmosphere during the year of sampling (Environment Canada, 2015). If dissolved $CO_2$ and
$CH_4$ concentrations in surface waters were above or below their corresponding calculated atmospheric equilibrium
concentrations, the freshwater systems were considered a source (+) or sink (-) relative to the atmosphere,
respectively.
We also measured ebullition fluxes of $CH_4$ to the atmosphere from two freshwater systems (Skeleton Lake,
Pond 01) during two growing seasons using manual bubble collection and GC analysis (see Supporting
Information).
**2.4 Supporting measurements**
We quantified additional physical and chemical parameters in surface waters at the same sites as we
collected our GHG samples, although at reduced sampling frequencies (Table S2). At each site, temperature, pH,
specific conductivity and dissolved $O_2$ were measured in-situ using a YSI (Yellow Springs, OH) 556 MPS multi-
probe. Water samples were also collected for general chemical analyses (total dissolved nitrogen [TDN], particulate
N, $NO_3^-+NO_2^-$, $NH_4^+$, total phosphorus [P], total dissolved P [TDP], alkalinity, dissolved organic carbon [DOC],
total dissolved solids, major cations/anions, chlorophyll-a [chl-a] into pre-cleaned HDPE bottles. These samples





were immediately processed in the Lake Hazen/Quttinirpaaq Field Laboratory clean room after water collection, and
stored in the dark at ~5ºC or frozen until analysed at the BASL.

**2.5 Net atmospheric exchange of $CO_2$ and $CH_4$ of a large high Arctic watershed**


To better understand the role of freshwater ecosystems in regional fluxes of carbon GHGs, freshwater $CO_2$
and $CH_4$ fluxes measured in this study were coupled with terrestrial fluxes measured in the watershed during the
2008-12 growing seasons (Emmerton et al., 2014; Emmerton et al., 2016). Areal coverage of the different ecosystem
types in the watershed was isolated from a previous classification of Quttinirpaaq National Park (Edlund, 1994)
using a Geographical Information System (ArcGIS v.10.3; ESRI, Redlands, US). Mean growing season fluxes from
each measured ecosystem were then weighted to matching coverage area in the watershed to estimate the total
carbon gas exchange with the atmosphere. Glacial ice was assumed to be a net-zero contributor of total watershed
gas exchange in this scaling exercise.

**3 Results**


**3.1 Biogeochemical classification of high Arctic ponds**


Four distinct types of freshwater systems were evident from our sampling in the Lake Hazen watershed
(Table 2; Figure 2; hierarchical cluster analysis; see Supporting Information). "Evaporative" ponds (Ponds 07, 10,
12) occurred in the upland of the Lake Hazen catchment and were hydrologically-isolated from their surrounding
basins post-snowmelt. These ponds were relatively high in concentrations of total dissolved solids, most measured
ions, DIC, DOC, organic particles, TDP and chl-a. Pond 03, though not technically clustered with others, was forced
to the Evaporative pond category based on lack of consistent inflowing water and high concentrations of most
dissolved ions. This delegation was further consistent with isotopic measurements of oxygen ($\delta^{18}O$-$H_2O$) in water
taken from each aquatic system in July 2010 (Figure S2). "Meltwater" systems, including Ponds 11, 16 and Skeleton
Lake, also occurred in the upland of the Lake Hazen watershed, but received consistent water supply through the
growing season primarily from snowmelt, permafrost thaw water and/or upstream lake drainage. The general
chemistry of these systems was therefore consistent and without extremes during the growing season. Typical
meltwater streams draining to these ponds were high in TDN and sulfate ($SO_4^{2-}$; Table 2). "Shoreline" ponds (Ponds
01, 02) occurred along the margin of Lake Hazen and were typically physically isolated from the large lake by
porous gravel berms, and surrounded by wetland soils and flora during spring low water conditions. As glacial melt



accelerated throughout the growing season, though, the water level of Lake Hazen rose and could seep through the
berms to incrementally flood the ponds and surrounding wetlands (Figure S3). Shoreline ponds changed chemically
during the onset of flooding as indicated, for example, by an increase in the concentration of reduced ions (i.e.,
$NH_4^+$; Table S4). A separate smaller cluster of Pond 01 samples occurred during particularly high-water periods
when Lake Hazen breached the berms (Figure 2). The flooding water from the "Lake Hazen shoreline" was cold,
dilute in dissolved ions, organic matter, TDN, and chl-a, but considerably higher in $NO_3^-$ compared to other water
bodies. A single sample from a pre-flooded Shoreline pond (Pond 02) grouped within the Lake Hazen shoreline
cluster likely because its water was ultimately sourced from Lake Hazen and because of its isolation from its
wetland margins during low-water conditions.
**3.2 Dissolved concentrations and net atmospheric exchange of $CO_2$ and $CH_4$ of high Arctic freshwaters**
**3.2.1 $CO_2$**
Growing season concentrations of dissolved $CO_2$ in sampled high Arctic freshwaters from 2005 to 2012
varied substantially within and between the system types, and therefore overall resulted in non-significant
differences between them (Figure 3, 4; linear mixed-model; α=0.05; see Supporting Information).
On average, Evaporative ponds had the highest mean $CO_2$ concentrations (mean±SE; 27.9±4.9 μmol $L^{-1}$)
compared to other pond types, primarily due to conditions in Pond 03 and Pond 07. These ponds were the shallowest
of the four sampled and were rich in reduced ions, DIC, DOC, total P and calcium. $CO_2$ concentrations were above
atmospheric equilibrium concentration and therefore these ponds were sources of the gas to the atmosphere
(+177±66 μmol $CO_2$ $m^{-2}$ $hr^{-1}$; Figure S4). The other Evaporative ponds (Ponds 10, 12) were deeper and had $CO_2$
concentrations that were typically near those of the atmosphere. This contributed to their near-zero exchange of $CO_2$
with the atmosphere (-5±17 μmol $CO_2$ $m^{-2}$ $hr^{-1}$). When combining all Evaporative ponds together, they were net
sources of $CO_2$ to the atmosphere (+73±93 μmol $CO_2$ $m^{-2}$ $hr^{-1}$).
Meltwater systems had lower, but insignificantly different, $CO_2$ concentrations (26.2±3.9 μmol $L^{-1}$) than
Evaporative ponds. Meltwater systems showed only gradual, venting-related declines of $CO_2$ concentrations through
the summer, with strong consistency in concentrations between sampling times and sites. However, they emitted
higher, though not significantly different, fluxes of $CO_2$ to the atmosphere overall (+160±66 μmol $m^{-2}$ $hr^{-1}$; Figure 4)
compared to the other types of systems. $CO_2$ concentrations of these systems correlated strongly and positively with
$CH_4$ concentrations, but negatively with other measurements that were of high concentrations in meltwater streams



draining into the systems (e.g., $SO_4^{2-}$, TDN; Table 2, S4). Mean diurnal trends in $CO_2$ concentrations across all
sampling years, as measured by the automated system at Skeleton Lake, showed that $CO_2$ and $O_2$ concentrations
associated positively together, but negatively with water temperature (Figure 5).
Mean $CO_2$ concentrations of Shoreline ponds (22.5±3.7 µmol $L^{-1}$) were similar to the other pond types,
which obscured their considerable seasonal changes within and between growing seasons. From 2005 to 2007, both
Pond 01 and Pond 02 received little floodwater from Lake Hazen due to lower lake water levels. These conditions
resulted in dense wetland vegetation growth surrounding the ponds and low mean daily dissolved $CO_2$
concentrations (6.5±0.4 µmol $L^{-1}$) and strong uptake of atmospheric $CO_2$ (-329±59 µmol $m^{-2}$ $hr^{-1}$). The drier wetland
state of these ponds changed in following summers when Lake Hazen rose substantially upon greater inputs of
glacial meltwaters (WSC, 2015), causing the rising waters to seep through porous berms into the ponds through
July. In concert with flooding, concentrations of $CO_2$ from 2008-11 of each pond together increased substantially
(30.1±1.5 µmol $L^{-1}$) resulting in strong net emissions of $CO_2$ to the atmosphere (+228±44 µmol $m^{-2}$ $hr^{-1}$). $CO_2$
concentrations of the ponds correlated strongly and positively with concentrations of many constituents in the same
waters (Table S4). Diurnal trends of $CO_2$ and $O_2$ concentration measured by the automated system at Pond 01 over
several growing seasons showed a primary production signature with opposite temporal patterns of the gases, with
greater $O_2$ during the warmest and lightest parts of the day (Figure 5). However, the net result of strong seasonality
in these ponds was slight net emission of $CO_2$ to the atmosphere (+42±60 µmol $m^{-2}$ hr) that was not statistically-
different from other types of systems.
Lake Hazen shoreline water, though not necessarily representative of the entire lake itself, was
characteristic of its moat occurring early each growing season, and of water that intruded Shoreline ponds in July.
This water was generally near atmospheric equilibrium concentrations of $CO_2$ (21.0±7.8 µmol $L^{-1}$) with stable and
low $CO_2$ uptake throughout the season (-44±66 µmol $m^{-2}$ hr). $CO_2$ concentrations of this shoreline water related
strongest and positively with DIC, major ions and wind speed (Table S4).
**3.2.2 $CH_4$**
Each of Evaporative, Meltwater and Lake Hazen shoreline freshwaters had statistically similar and low
$CH_4$ concentrations (0.06-0.14 µmol $L^{-1}$) and fluxes (0-3 µmol $m^{-2}$ $hr^{-1}$) across all growing seasons (Figure 3,4, S4).
Evaporative ponds had generally flat seasonal $CH_4$ concentration and flux trends, except for an outlier sample from
Pond 10 in mid July 2011. Meltwater systems were also generally low in $CH_4$ concentrations and fluxes through the



summers and associated strongly with similar chemical measures as $CO_2$ (Table S4). Notable flux emissions from
these systems only occurred during episodic wind events, also similar to $CO_2$ (Figure S4). However, unlike $CO_2$,
higher $CH_4$ concentrations were sustained into July in Skeleton Lake in 2010. Lake Hazen shoreline water showed
low and stable $CH_4$ concentrations and fluxes each growing season with infrequent and small releases of the gas to
the atmosphere. $CH_4$ concentrations in this water correlated positively and strongly with particulate carbon
concentrations (Table S4).

Shoreline ponds, alternatively, had significantly higher $CH_4$ concentrations relative to the other systems

($1.18\pm0.16$ µmol $L^{-1}$) and showed a dynamic seasonal pattern dominated by the timing of flooding. In 2005 and
2007 before substantial seasonal flooding started to occur, $CH_4$ concentrations ($0.29\pm0.03$ µmol $L^{-1}$) and fluxes to
the atmosphere ($8\pm2$ µmol $m^{-2}$ $hr^{-1}$) were low. As the Shoreline ponds began to receive $NO_3^-$-rich flood water from
Lake Hazen by mid-summer in subsequent years (Table S4), 2008-11 $CH_4$ concentrations and fluxes increased
substantially ($1.70\pm0.13$ µmol $L^{-1}$; $41\pm10$ µmol $m^{-2}$ $hr^{-1}$). This significant increase in $CH_4$ flux emissions from
Shoreline ponds during flooding (>five times higher than during dry periods) was coupled with large increases in
pond surface areas, effectively producing even higher total $CH_4$ emissions to the atmosphere. Towards the end of
July during flooding conditions, full berm breach of the Shoreline ponds by rising Lake Hazen waters occurred
resulting in rapid dilution of $CH_4$ concentrations, but logistical constraints prevented later summer sampling to
investigate if concentrations rebounded thereafter. Overall, aided by poor solubility of $CH_4$ in water and episodic
wind events (Figure S4), the flooding of Shoreline ponds drove significantly larger $CH_4$ emissions to the atmosphere
compared to other pond types ($+28\pm5$ µmol $m^{-2}$ $hr^{-1}$; Figure 4).

**3.3 Net atmospheric exchange of $CO_2$ and $CH_4$ of a large high Arctic watershed**

Emmerton et al. (2014; 2016) measured, using eddy covariance flux towers ($CO_2$, $CH_4$) and static chambers

($CH_4$), growing season carbon GHG exchange with terrestrial polar semidesert and meadow wetland landscapes
from 2008-12. They found that the dry and mostly barren polar semidesert was among the most unproductive
terrestrial ecosystems on Earth, taking up only -0.004 g C-$CO_2$ $m^{-2}$ $d^{-1}$ during the growing season, similar to findings
from other studies (Soegaard et al., 2000; Lloyd, 2001; Lund et al., 2012). When scaled to total watershed area
including Lake Hazen (7,443 $km^2$), polar semidesert landscapes were inconsequential to total $CO_2$ exchange (-1,253
Mg C-$CO_2$; 9% of total exchange) despite comprising a substantial proportion of the catchment (3,819 $km^2$; 51%;
Figure 6). All types of standing freshwaters sampled in the watershed from this study showed statistically-similar



$CO_2$ fluxes compared to the polar semidesert. When assuming its shoreline waters were representative of the entire
lake area, the expansive Lake Hazen (542 $km^2$; 7%) exchanged relatively little $CO_2$ with the atmosphere (-721 Mg
C-$CO_2$; 5%), as did smaller freshwater systems (145 $km^2$; 2%) in the watershed (+600 Mg C-$CO_2$; 4%). In clear
contrast, during the growing season, moist and vegetated meadow wetland ecosystems were found to consume $CO_2$
at rates similar to wetlands in the southern Arctic (-0.96 g C-$CO_2$ $m^{-2}$ $d^{-1}$). Consequently, meadow wetlands
exchanged an estimated 82% (-11,368 Mg C-$CO_2$) of total $CO_2$ with the atmosphere despite occupying only 2%
(129 $km^2$) of the area in the Lake Hazen watershed. Total $CO_2$ exchange of the watershed was -10,236 Mg C-$CO_2$ (-
1.375 g C-$CO_2$ $m^{-2}$) during the growing season.

The high Arctic polar semidesert has recently gained attention as a notable atmospheric sink of $CH_4$ (-0.001

g C-$CH_4$ $m^{-2}$ $d^{-1}$; Emmerton et al., 2014), which has since been observed in studies at other high Arctic locations
(e.g., Jorgensen et al., 2015). These uptake fluxes coupled with its expansive coverage made the polar semidesert the
key landscape controlling net $CH_4$ exchange throughout the Lake Hazen watershed (-412 Mg C-$CH_4$; 94% of total
exchange; Figure 6). Surprisingly, a productive meadow wetland in the watershed was a weaker emitter of $CH_4$ to
the atmosphere (+0.001 g C-$CH_4$ $m^{-2}$ $d^{-1}$) compared to other high Arctic wetlands (Emmerton et al., 2014), releasing
only 10 Mg C-$CH_4$ (2%) to the atmosphere during the growing season. All upland freshwater systems (Evaporative
+ Meltwater) had low emissions of $CH_4$ to the atmosphere (11 Mg C-$CH_4$; 2%), as did Lake Hazen itself (+6 Mg C-
$CH_4$; 1%). All measured ecosystems had statistically-similar $CH_4$ fluxes except for the strong $CH_4$-producing
Shoreline ponds (Figure 6). However, poor areal coverage of these dynamic systems in the watershed (0.6 $km^2$;
<1%) resulted in contributions of <<1% (+0.4 Mg C-$CH_4$) of all $CH_4$ exchange in the Lake Hazen watershed (-385
Mg C-$CH_4$; -0.052 g C-$CH_4$ $m^{-2}$).
**4 Discussion**
**4.1 Dissolved concentrations and net atmospheric exchange of $CO_2$ and $CH_4$ of high Arctic freshwaters**
**4.1.1 $CO_2$**

Concentrations of $CO_2$ and other compounds were highest in small and shallow Evaporative ponds (Ponds

03, 07) compared with those that were larger and deeper (Ponds 10, 12). Dissolved $CO_2$ was likely being produced
effectively in all Evaporative ponds by considerable ecosystem metabolism and accumulation and dissociation of
weathered carbonates and evaporates (Trettin, 1994; Marcé et al., 2015). However, $CO_2$ was likely more effectively





diluted in the larger ponds and therefore less susceptible to wind-related turbulence and gas exchange with the
atmosphere. Meltwater systems showed steady biogeochemical conditions but similar $CO_2$ concentrations as other
freshwater types, despite inclusion of early summer sampling at Skeleton Lake (2007, 2010). High $CO_2$
concentrations in Skeleton Lake during that time were typical of post-ice covered waters only beginning to re-
equilibrate with the atmosphere (Kling et al., 1992; Karlsson et al., 2013). Greater exchange of $CO_2$ by Meltwater
systems, however, was not primarily driven by early season venting or sustained exchanges compared to other
ponds, but rather by higher frequency of episodic releases of $CO_2$ to the atmosphere (Figure S4). This may have
been related to their greater mean depths, which promoted stratification in at least one of our sampled Meltwater
systems (Skeleton Lake; Figure S5). Stratification would confine decomposition products (e.g., $CO_2$, $CH_4$) to near
their sites of origin in bottom sediments and extensive benthic mat communities (Rautio et al., 2011), which would
then be released most readily during and after wind mixing events. We observed evidence of this process via strong
positive associations between $CO_2$ and $CH_4$ concentrations in surface waters (Table S4). Further, mean diurnal $CO_2$
and $O_2$ concentrations in surface waters trended similarly with temperature- or wind-related solubility changes,
rather than oppositely if metabolic processes (i.e., primary productivity or decomposition of organic matter) were
important drivers in surface waters. Shoreline ponds changed drastically in size and chemistry in response to
seasonal flooding by Lake Hazen shoreline water. During pre-flooding conditions, $CO_2$ concentrations were low
which could be attributed to DIC use by autotrophic plankton (pre- to post-flooding mean chl-a concentrations of 1.2
to 0.4 μg $L^{-1}$), but more likely by observed dense benthic and macrophytic communities along the margins of the
ponds. When inundated by flood waters, $CO_2$ concentrations rose sharply which is typically observed in flooded
wetlands (Kelly et al., 1997). This occurs because widespread inundation of plants and soils typically prompts rapid
decomposition and propagation of reduced compounds (e.g., $NH_4^+$; Table S4). Although diurnal $CO_2$ and $O_2$
concentrations suggest that primary productivity was consistently occurring in Shoreline pond surface waters, this
pattern appeared overwhelmed by acute seasonal adjustments in $CO_2$ exchange driven by flooding.

$CO_2$ concentrations in Lake Hazen shoreline water were near atmospheric equilibrium and only weakly

consumed atmospheric $CO_2$. These results along the shoreline appear to be similar to other locations offshore
(unpublished) and were reflective of most deep lakes with extremely low nutrient, organic matter and chl-a
concentrations (0.20 μg $L^{-1}$; Keatley et al., 2007; Babaluk et al., 2009). $CO_2$ gas exchange between the lake and the
atmosphere correlated well with DIC, alkalinity and other ions, which suggests supply and dissociation of carbonate



material from the watershed, as well as wind mixing, were important factors contributing to Lake Hazen surface
water $CO_2$ concentrations, rather than primary productivity or heterotrophic decomposition.
**4.1.2 $CH_4$**
Evaporative and Meltwater systems were typically weak producers and emitters of $CH_4$, which was
possibly sustained by concurrently high $SO_4^{2-}$ concentrations in these systems (Table 2; Trettin, 1994). This may
have given competitive advantage to $SO_4^{2-}$-reducing bacterial communities in sediments, which typically
outcompete methanogenic bacteria for hydrogen. This hypothesis was supported by the prevalence of $H_2S$ gas in
collected sediment cores from Skeleton Lake (unpublished) and by the trivial fluxes of $CH_4$ in bubbles measured
emerging from sediments (0.00-0.01 mg m$^{-2}$ d$^{-1}$; Table S5; see Supporting Information). Low production and
exchange of $CH_4$ in Lake Hazen, alternatively, was most likely associated with the lake's ultra-oligotrophic
standing, well-oxygenated water, and little accumulation of littoral organic matter where anoxia could prevail and
$CH_4$ be produced. Only during periods of strong wind mixing of surface waters, or when Shoreline ponds breached
and released particulate organic matter, did $CH_4$ release from the shoreline of the lake to the atmosphere increase
above near-zero values.
Shoreline ponds were regional "hot-spots" of $CH_4$ exchange, which was clearly driven by seasonal
flooding, similar to that described for $CO_2$ exchange. Pre-flooding conditions in the ponds were characterized by dry
and oxygenated wetland soils which were exposed to the atmosphere and not connected to the central pond where
we sampled. Flooding induced saturation of organic soils surrounding the wetland and perhaps provided
advantageous conditions for anaerobic metabolism, including methanogenesis. This may have been further
supported by the flushing of the ponds with $SO_4^{2-}$-poor Lake Hazen water, therefore favoring metabolism of
methanogens over $SO_4^{2-}$-reducers in the flooded soils.
**4.3 Net atmospheric exchange of $CO_2$ and $CH_4$ of a large high Arctic watershed**
Most studies of terrestrial and freshwater carbon GHG exchange in the broader Arctic do not occur
concurrently at a specific location. Our multiple season program of measuring carbon GHG exchange of both
freshwater and terrestrial ecosystems provided a unique opportunity to delineate the relative strengths of freshwater
and terrestrial contributions to regional carbon cycling in a high Arctic watershed (Table 3). Polar semideserts are
typically dry and barren landscapes with little vegetation growth or organic matter and nutrient accumulation to



drive atmospheric exchange of $CO_2$. Not surprisingly, freshwaters receiving runoff from the polar semidesert also
support mostly underproductive ecosystems which exchange little $CO_2$ with the atmosphere (Figure 6). Lake Hazen,
similarly, is sustained by cold and sediment-laden glacial melt water limited in compounds essential for life.
Together, these oligotrophic freshwater and terrestrial ecosystems largely characterize the current low-production
state of the Lake Hazen watershed and much of the high Arctic in general. Meadow wetlands, alternatively, are
topographical lowlands with flowing water which are ideal high Arctic environments for vegetation growth and
accumulation of soils rich in organic matter and nutrients. Conditions in these spatially-limited meadow wetlands
are so profound relative to other ecosystem types in the Lake Hazen watershed that they currently drive a strong
majority of atmospheric $CO_2$ exchange there. Evidently, $CO_2$ uptake across this, and likely other, high Arctic
landscapes is inherently tied to the availability and movement of water.

Though polar semidesert landscapes exchanged only limited amounts of $CO_2$ with the atmosphere, they

were extremely important sinks of atmospheric $CH_4$, such that they dominate high Arctic $CH_4$ cycling. This strong
sink has been attributed to soil conditions that promote efficient atmospheric gas diffusion and temperature
sensitivity of methanotrophic bacteria at high latitudes (Emmerton et al., 2014; Jorgensen et al., 2015). Any
ecosystems where water saturation and anoxia would be expected to prevail during the growing season (i.e., small
ponds, lakes and meadow wetlands) were surprisingly weak emitters of $CH_4$. This may have been due to poor
quantities of organic substrates in the soils or beds of these systems, oxidation zones near gas exchange sites, or
biogeochemical constraints related to $SO_4^{2-}$. Regardless, $CH_4$ exchange in high Arctic ecosystems, at least those
similar to the landscape composition of the Lake Hazen watershed, cannot be overlooked as substantial sinks of this
potent GHG. However, these conditions, which currently define carbon GHG exchange in the Lake Hazen
watershed, may be poised to change.

Warming growing season conditions in the Lake Hazen watershed have affected temperature sensitive

components of the landscape including deep soils and permafrost (unpublished). Warming has also affected the
region's hydrology through the greater delivery of glacial melt water to Lake Hazen and increased frequency and
extent of ice-free area across the lake. Further, other studies suggest that changing air masses and evaporation from
newly exposed coastal waters due to sea ice loss will deliver increased precipitation to high Arctic landscapes
(Bintanja and Selten, 2014). These intensifying temperature and precipitation conditions will likely accelerate
landscape changes in the watershed, especially considering the contemporary climate in the region is one at the low



global extreme of summer air temperatures and water availability. For example, Evaporative ponds could expand
and deepen with greater precipitation and snow runoff, possibly repressing $CO_2$ emissions similar to Ponds 10 and
12. Greater heating without sustained precipitation may alternatively cause these systems to continue to shallow and
possibly strengthen GHG emissions to the atmosphere, similar to Ponds 03 and 07, but would be susceptible to
ultimate drying (Smol & Douglas, 2007). Meltwater systems, alternatively, may be resistant to climate-related
influences because of their already steady water supply and greater volumes, though future water delivery may be
affected by diminishing stores of permafrost ice in soils. Shoreline ponds ultimately may endure the most substantial
changes in GHG emissions of high Arctic water bodies. Lake Hazen, under warming and wetting conditions, would
be expected to receive greater amounts of glacial melt water runoff during the growing seasons. Earlier rises in Lake
Hazen water levels would cause earlier flooding of Shoreline ponds and sustain longer periods of organic soil and
vegetation inundation, resulting in larger $CO_2$ and $CH_4$ emissions to the atmosphere. Eventually, though, extensive
flooding may prevent the macrophyte growth currently found in these Shoreline ponds, decreasing the amount of
fresh organic available for decomposition once the flooding occurs. However, based on areal coverage in the
watershed, future changes in the GHG exchange of the region is likely dependent on changes occurring at the polar
semidesert and in Lake Hazen itself. Polar semideserts and meadow wetlands represent a gradient of productivity
defined by increasing water availability. If increased heating and precipitation results in greater soil retention of
water and support for greater summer vegetation, then the sink strength of $CO_2$ in the high Arctic may improve
drastically, but may at the same time reduce atmospheric $CH_4$ oxidation in soils. However topographical constraints
of water flow may limit the magnitude of vegetation growth in meadow wetlands and across the polar semidesert in
the shorter term. For Lake Hazen itself, we may expect that a warming climate will hasten loss of ice from the lake,
resulting in greater water column heating, longer growing seasons, and increased river inflow to the lake. This may
induce more intense mixing and nutrient availability in the water column and perhaps higher productivity and $CO_2$
sequestration. Loss of glacial ice also presents new, but uncertain, opportunities of gas exchange on newly exposed
landscapes.
Ultimately, freshwater systems in the Lake Hazen watershed did not produce considerable enough fluxes of
GHGs relative to the atmosphere to dominate regional GHG exchange, as observed in other studies to the south.
However, potential does exist in the watershed for hot-spots of GHG exchange to emerge from greater water
availability, which is a distinct possibility in the near-future with ongoing rapid climate warming in the region.





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

**Acknowledgements**

This work was supported by the Natural Sciences and Engineering Research Council (Discovery Grants
Program, Northern Research Supplement, Canada Graduate Scholarship), Natural Resources Canada (Polar
Continental Shelf Project), the Canadian International Polar Year (Climate Change Impacts on Canadian Arctic
Tundra project; Government of Canada), the Canadian Circumpolar Institute (Circumpolar Boreal Alberta Research
grant), the Association of Canadian Universities for Northern Studies (ACUNS), ArcticNet, and Indigenous and
Northern Affairs Canada (INAC; Northern Scientific Training Program). We are grateful for the logistical, technical
and field support of Parks Canada at Quttinirpaaq National Park and the Polar Continental Shelf Project–Resolute
Bay. We thank Dr. Sherry Schiff and Dr. Jason Venkiteswaran for providing isotopic analyses. We are also thankful
of the tremendous support provided by the University of Alberta Biogeochemical Analytical Service Laboratory,
Claude Labine and Campbell Scientific Canada Corp., Elizabeth (Ela) Rydz, Hayley Kosolofski, Patrick Buat, Dr.
Micheline Manseau and Dr. Raymond Hesslein.





**Tables**
**Table 1 Morphometry and hydrology of ponds and lakes sampled for greenhouse gases concentrations and general**
**chemistry in the Lake Hazen (LH) watershed during the growing seasons of 2005, and 2007-2012.**

| Lake or Pond (location) | | Surface area (ha) | Mean depth (m) | Max. depth (m) | Elevation (m asl) | Primary water sources |
|---|---|---|---|---|---|---|
| Pond 01 | (N81.822 W71.352) | 0.2-0.7 | 0.2-0.6 | 0.5-1.3 | 166 | LH, snowmelt |
| Pond 02 | (N81.811W71.453) | 0.2-3.4 | 0.1-0.4 | 0.3-1.2 | 165 | LH, snowmelt |
| Pond 03 | (N81.829 W71.462) | 0.04 | 0.3 | 0.8 | 338 | Snowmelt |
| Pond 07 | (N81.835 W71.305) | 0.4 | 0.1 | 0.3 | 184 | Snowmelt |
| Pond 10 | (N81.838 W71.343) | 2.5 | 1.1 | 2.4 | 222 | Snowmelt |
| Pond 11 | (N81.832W71.466) | 0.2 | 1.1 | 2.5 | 291 | Snowmelt, permafrost |
| Pond 12 | (N81.831W71.529) | 0.2 | 0.8 | 1.9 | 370 | Snowmelt |
| Pond 16 | (N81.850W71.392) | 0.8 | 1.1 | 2.1 | 434 | Snowmelt, permafrost |
| Skeleton L. | (N81.829 W71.480) | 1.9 | 1.9 | 4.7 | 299 | Snowmelt, permafrost |
| LH-shore | (N81.821 W71.352) | 54,200 | 95[a] | 267[a] | 158 | Glacial, snowmelt |

[a]*Kock et al., 2012*




**Table 2 Mean water temperature and general chemistry of different freshwater types in the Lake Hazen (LH) watershed**
**during the growing seasons of 2005, 2007-2012. All measurements are in µmol L$^{-1}$ except for W$_T$ (°C), TDS (mg L$^{-1}$) and**
**chl-*a* (µg L$^{-1}$).**

| Lake Type | W$_T$ | pH | TDS | PC | DIC | DOC | NO$_3^-$ | NH$_4^+$ | TDP | Fe | SO$_4^{2-}$ | Chl-a |
|---|---|---|---|---|---|---|---|---|---|---|---|---|
| Evaporative | 10.8 | 8.3 | 1,020 | 49 | 2,122 | 2,374 | 0.01 | 0.5 | 0.32 | 1.13 | 5,320 | 1.3 |
| Meltwater | 11.0 | 8.2 | 333 | 23 | 1,458 | 452 | 0.02 | 1.9 | 0.17 | 0.05 | 1,755 | 0.5 |
| Shoreline | 11.6 | 8.2 | 173 | 32 | 1,691 | 311 | 0.11 | 2.0 | 0.18 | 1.56 | 365 | 0.4 |
| LH-shoreline | 5.4 | 7.9 | 59 | 10 | 524 | 51 | 0.24 | 1.8 | 0.08 | 0.04 | 69 | 0.1 |
| Meltwater source | 2.6 | 7.6 | 653 | - | 769 | 67 | 7.70 | 0.1 | 0.05 | 0.6 | 3318 | - |


*W$_T$: water temperature; TDS: total dissolved solids; PC: particulate carbon; DIC: dissolved inorganic carbon; DOC: dissolved*
*organic carbon; NO$_3^-$: dissolved nitrate + nitrite; NH$_4^+$: dissolved ammonium; TDP: total dissolved phosphorus; Fe: dissolved*
*iron; SO$_4^{2-}$: dissolved sulfate; chl-a: chlorophyll-a*





**Table 3 Ranges or means of $CO_2$ and $CH_4$ fluxes (g C m$^{-2}$ d$^{-1}$) from selected studies investigating terrestrial and**
**freshwater greenhouse gases exchange during the growing season of high, low and subarctic regions. Positive values**
**represent emission to the atmosphere.**

| Location* | Ecosystem | Period | $CO_2$ flux | $CH_4$ flux | Study |
|---|---|---|---|---|---|
| **High Arctic** | | | | | |
| Lake Hazen, CA | Dry tundra | Jun.-Aug. | -0.00 | -0.00 | a, b |
| Zackenberg, GR | Heath | May-Sep. | -0.39,0.03 | 0.04,0.06 | c, d |
| Lake Hazen, CA | Ponds | Jun.-Aug. | -0.01,0.05 | 0.00,0.01 | This study |
| Pond Inlet, CA | Ponds | Jul. | -0.22,0.72 | 0.00,0.07 | e |
| **Low Arctic** | | | | | |
| Lena delta, RU | Wet tundra | May-Aug. | -0.35 | 0.01 | f, g |
| Barrow, US | Moist tundra | Jun.-Sep. | -0.02,0.66 | 0.02 | h, i |
| Lena delta, RU | Ponds | Aug.-Sep. | 0.38,1.10 | | j |
| Toolik Lake, US | Lakes | Jul.-Aug. | -0.07,0.72 | 0.96,12.25 | k |
| Yukon delta, US | Lakes | Jun.-Aug. | | 0.04 | l |
| **Subarctic** | | | | | |
| Chokurdakh, RU | Tundra | Jul-Aug | | 0.05 | m |
| Abisco, SE | Shrub tundra | Jul-Aug. | -0.95,-0.83 | | n |
| Cherskii,, RU | Tussock tundra | Jul.-Aug. | -0.15,0.50 | 0.32 | o |
| James Bay, CA | Mixed tundra | Jun-Oct. | | 0.04 | p |
| Narvik, SE | Lakes | May-Oct. | -0.03,0.13 | 0.00,0.02 | q |
| Inuvik, CA | Lakes | Jun.-Aug. | -1.55,-0.65 | - | r |
| Cherskii, RU | Ponds | Jun.-Sep. | | 0.09 | s |


*a Emmerton et al., 2016; b Emmerton et al., 2014; c Lloyd, 2001; dTagesson et al., 2012; e Laurion et al., 2010; f Kutzbach et*
*al., 2007; g Sachs et al., 2010; h Kwon et al., 2006; i Sturtevant and Oechel, 2013; j Abnizova et al., 2012; k Kling et al., 1992; l*
*Fan et al., 1992; m Parmentier et al., 2011; n Fox et al., 2008; o Merbold et al., 2009; p Roulet et al., 1994; q Karlsson et al.,*
*2013; r Tank et al., 2009; s Walter et al., 2006; *as delineated by AMAP, 1998.*





**Figures**

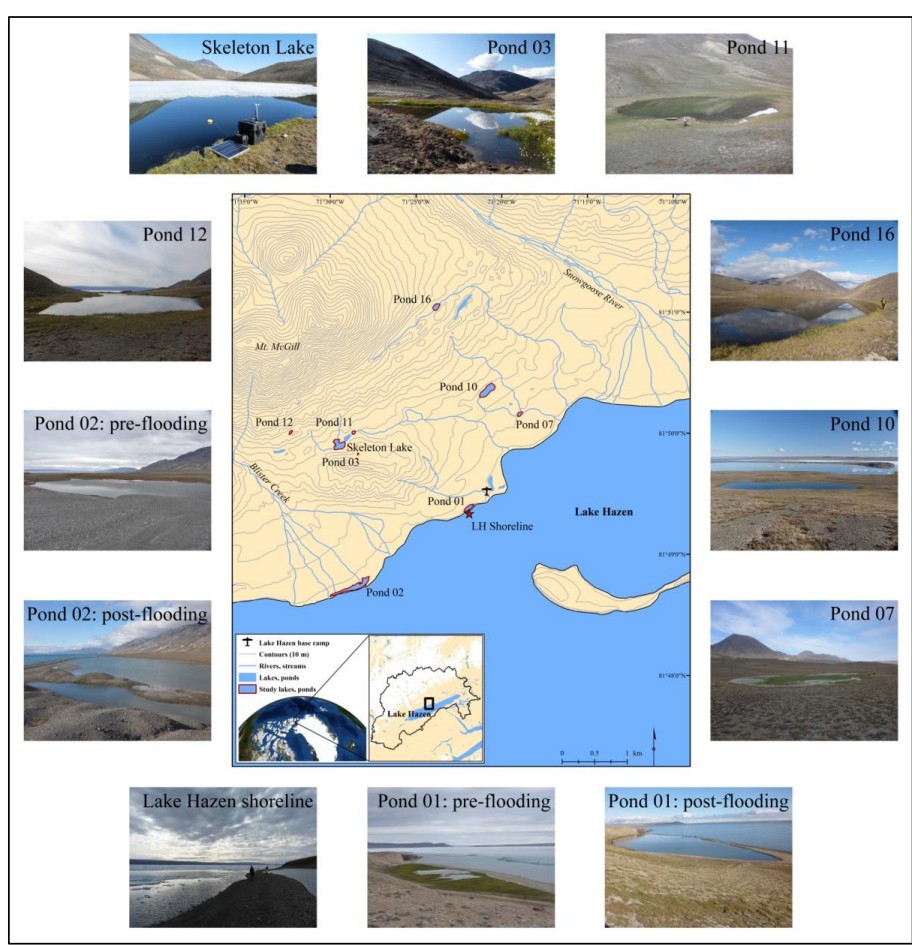

**Figure 1 Map of the Lake Hazen base camp in Quttinirpaaq National Park, Nunavut, Canada. Ponds and lakes investigated in this study are indicated on the map and shown in photographs. Shown inset of the map are the general locators of the Lake Hazen watershed.**




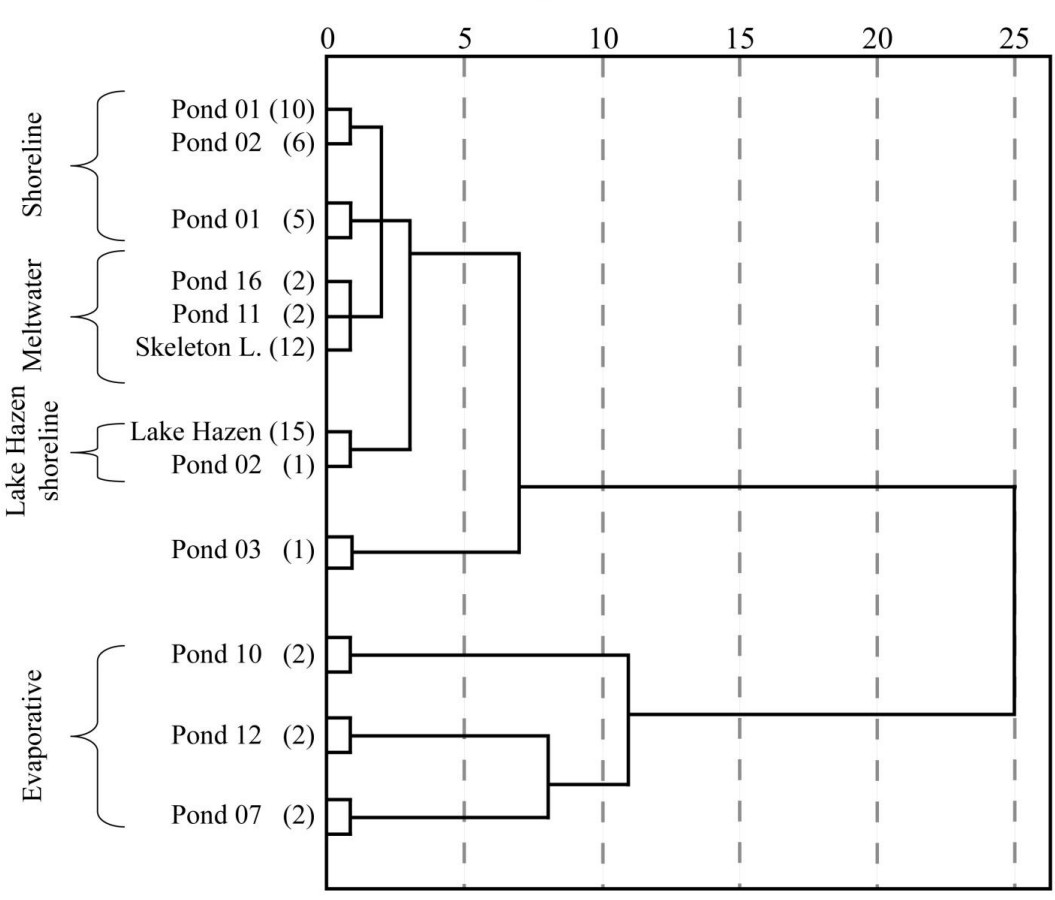


**Figure 2 Dendogram of sampled high Arctic freshwater systems in the Lake Hazen watershed between 2005, and 2007-**
**2012 (hierarchical cluster analysis; see Supporting Information). Water chemistry (see Methods) and carbon greenhouse**
**gases concentrations measured periodically from 10 locations (Figure 1) were used as inputs to the analysis.**



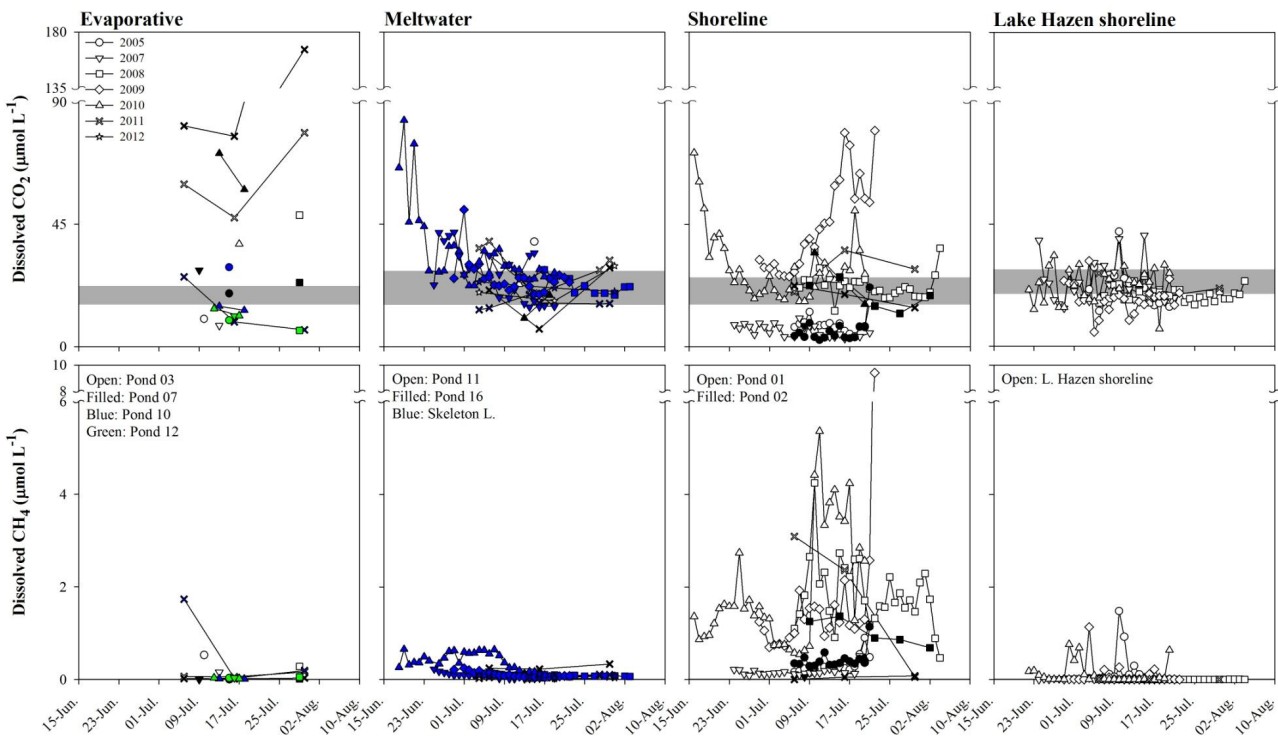


**Figure 3 Dissolved carbon dioxide ($CO_2$) and methane ($CH_4$) concentrations during the 2005, and 2007-2012 growing seasons (June-August) from different types of high Arctic freshwater systems in the Lake Hazen watershed. Inset text shows site names within each freshwater type. Grey areas indicate the range of atmospheric equilibrium concentrations during the sampling period.**




**Figure 4 Mean (±SE) dissolved carbon dioxide (CO$_2$) and methane (CH$_4$) concentrations and fluxes during the 2005, and 2007-2012 growing seasons from four different freshwater system types in the Lake Hazen watershed. Letters denote statistical differences between ecosystem types for each gas (linear mixed model; α=0.05; see Supporting Information).**





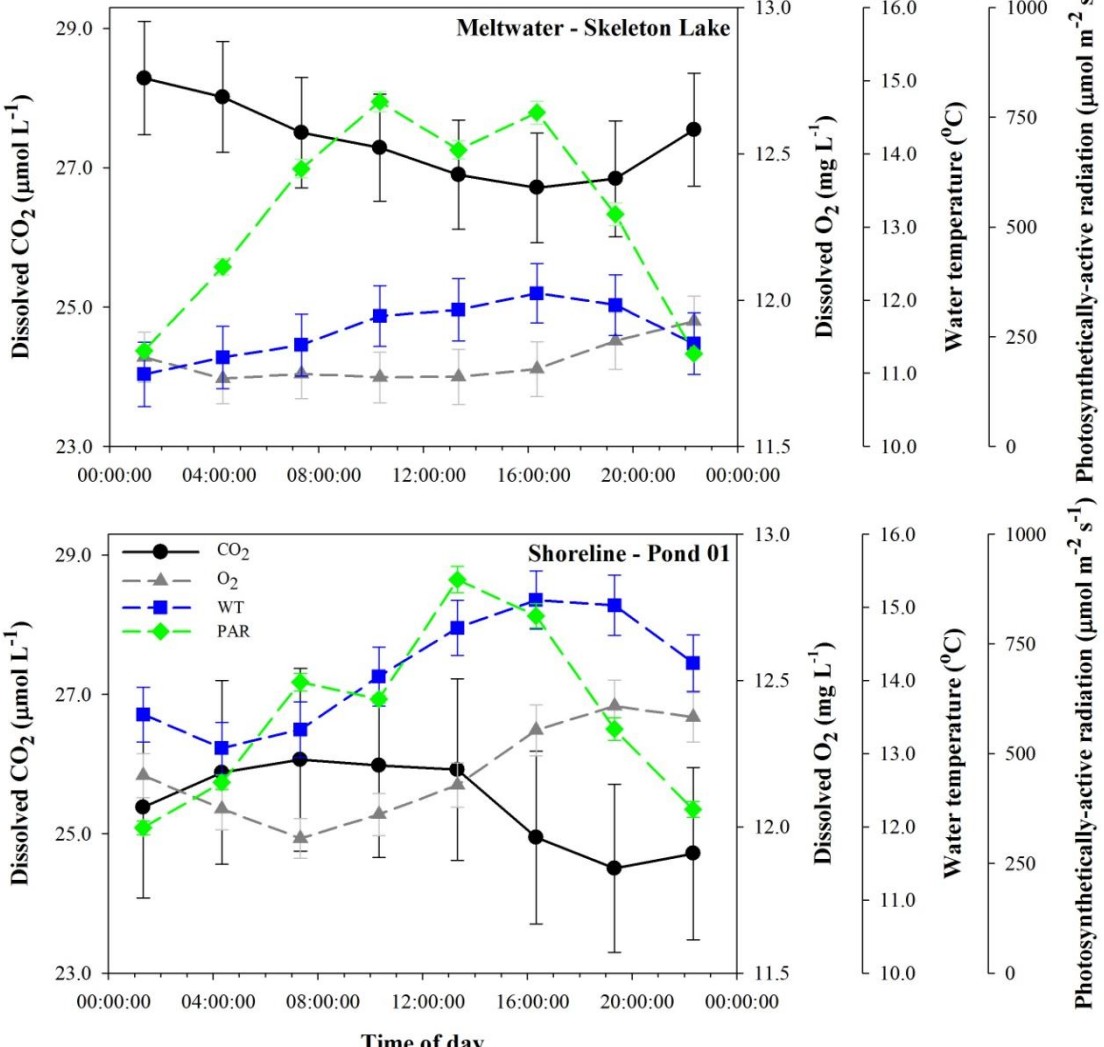

**Figure 5 Three-hour diurnal dissolved carbon dioxide (CO2) concentration, oxygen (O2) concentration, water temperature and photosynthetically-active radiation (PAR) data measured by automated systems deployed at the shorelines of Skeleton Lake (2008-10) and Pond 01 (2008-10) during the high Arctic growing season in the Lake Hazen watershed.**





**Figure 6 (a.) Comparison of the net exchange of carbon dioxide ($CO_2$) and methane ($CH_4$) between high Arctic terrestrial and freshwater ecosystems and the atmosphere in the Lake Hazen watershed during the growing seasons of 2005, and 2007-2012. Letters denote statistical differences between ecosystem types for each gas (linear mixed model; $\alpha=0.05$; see Supporting Information). (b.) Total growing season (June-August) watershed exchange of $CO_2$ and $CH_4$ by terrestrial and freshwater types in the total Lake Hazen watershed.**