# Peer review of "The importance of freshwater systems to the net atmospheric exchange of carbon dioxide and methane with a rapidly changing high Arctic watershed"

_Biogeosciences, 2016_

## Referee Comment (RC1) · Anonymous Referee #1 · 15 Jun 2016

General comments on the overall quality of the paper

In this study, authors examine CO2 and CH4 exchange between four common freshwater systems located in the high Arctic watershed of Lake Hazen, Elsmere Island, Nunavut, Canada. This study aims to measure net exchange of CO2 and CH4 between atmosphere and common high Arctic freshwater ecosystems. In previous studies, authors investigated CO2 and CH4 net exchanges from terrestrial ecosystems (e.g. polar semi-desert, meadow wetlands, uplands) in the watershed of Lake Hazen. In this study, authors also aim to contextualize their new findings about freshwater systems in Lake

[Figure]

Hazen watershed with both their previous results and literature.

The scientific context and questions are well defined but specific issues about High Arctic landscapes could be clarified. What are the conditions and issues in High Arctic environments that differ from Low Arctic and Sub Arctic regions? The objectives are well exposed and correspond to data showed by authors. However, because of too general sentences and a lack of references, the answer of the second objective (contextualization) is not well developed and not accurate enough.

The authors present interesting data about $CO_2$ and $CH_4$ concentrations and exchange with atmosphere in freshwater systems in a high Arctic watershed. Measurements have been performed during summers from 2005 to 2012 in four freshwater ecosystems: evaporative ponds, meltwater ponds, shoreline ponds and Lake Hazen shoreline. These four freshwater systems were identified using a hierarchical clustering analysis based on gas concentrations and biogeochemistry data.

The main findings from this study are: Mean $CO_2$ concentrations and atmospheric exchanges were statistically similar between freshwater systems. The three types of ponds were weak $CO_2$ emitters. Shoreline ponds exhibited the highest dissolved $CH_4$ concentrations and fluxes to the atmosphere. However, because shoreline ponds cover a small area, their contribution to overall $CH_4$ emissions was weak. Other ponds and the Lake Hazen shoreline showed similar $CH_4$ concentrations and fluxes, which were weak. The same authors evidenced in previous studies that polar semi-desert ecosystems were weak sink of atmospheric $CO_2$ but they significantly consumed $CH_4$. Alternatively, meadow wetlands were important sink of $CO_2$ and weak emitter of $CH_4$. Considering their cover surface, all freshwater systems did not significantly contribute to total net C exchange from Lake Hazen watershed.

This study showed interesting results about the weak $CO_2$ and $CH_4$ emissions and uptake from freshwater systems in High Arctic region. These results are crucial to better constrain the assessment of future carbon feedback from permafrost environments

with climate change. Furthermore, high values of CH4 concentrations and fluxes in shoreline ponds are important considering likely evolution of both the water level and the biogeochemistry of Arctic lakes.

From my point of view, the main result is the seasonal flooding of Lake Hazen that led to strong increase in CO2 and CH4 emissions from ponds bordering the lake while other ecosystems were weak CH4 emitter. The highlighted processes are interesting and important although more evidences of the impact of biogeochemistry change on CH4 emissions would be necessary.

The second part of the discussion infers about the evolution of carbon exchange from freshwater systems in warmer and wetter conditions due to climate change. This section (paragraph 4.3) should be strongly modified. The current discussion section is too general and not adequately based on the findings from this study. There is a strong lack of references in this section (only four, an one auto-citation). Authors should develop a more specific and accurate discussion using more references.

Despite the importance of data, authors are strongly recommended to do a major revision before acceptation by 1) better investigating the biogeochemical processes responsible of contrasted CO2 and CH4 concentrations and fluxes among ponds and 2) strongly improving the discussion.

Scientific questions and issues

- I would recommend changing the title that does not reflect the main findings

- The number of samples should be indicated. The standard deviation of the mean should be indicated. How are representative the different measurements considering the differences in quantity of samples? How evolved the number of samples during time from 2005 to 2012? How many samples per site did you use to build the dendrogram?

- Are the ponds permanent throughout the year? Do you consider these freshwater systems as ponds of small lakes?

- What is the geologic substrate and the soil nature in the watershed? It may help to discuss your interesting results.

- Place the section 'numerical analysis' currently located in the supplementary data in the main manuscript.

- Authors studied both spatial and temporal variability. The two perspectives are not clearly exposed. I would recommend separating results about spatial variability from temporal evolution of gas concentrations and fluxes. The robustness of the spatial variability should be better explained by improving Figure 2 and Table 2. The available samples/data and the significance of differences in biogeochemical composition should be added. Some temporal trends should be better illustrated and explained (Figure 3).

- This manuscript reproduces and repeats some results already presented in Emmerton et al. (2014). Results from previous studies should be removed from the abstract and from the result section.

- Most of the figures should be modified in order to clarify the main information. Concentrations and fluxes bar plots should be separated; vertical scales should be changed. Some figures in supplementary data could be placed in the main manuscript such as Figure S2.

- Authors highlighted interesting biogeochemical processes, which could be better evidenced.

- This manuscript requires a substantial improvement of the section 4.3. Scientific arguments should be more specific and based on the findings from this study.

Technical corrections

In the introduction:

From l 35-36: The paragraph seems to be general and does not provide precise information about the weight of each process or where do they mainly occur? Is there a

latitudinal gradient from Sub-Arctic to Low-Arctic and High-Arctic.

l 41-42: Check sentence structure

l 45-46 and l51: The freshwater systems cover more than 50% of area in northern regions but less than 5% in polar semi-desert landscapes? Authors may better explain this important difference between the general point of view and specific semi-desert landscapes, and could describe the latitudinal/landscape gradient?

l 61-62: Control the relevance of chosen references, particularly Peterson et al. (2002) about permafrost thaw and Manabe et al., (1994) about growing seasons.

l 68: the reference (Antony et al., 2014) does not correspond to the sentence about polar semidesert.

l 68-70: Sentence is not clear, check its structure

l 70: although it is uncertain how rapid climate change will alter the C cycle in northern landscapes, this study does not provide strong information about its evolution.

In Methods

Figure 1: the figure should be modified. The general maps are too small and thus not useful, the north arrow and the scale are also too small and not visible. Even on a half page, pond and lake pictures are small and don not provide any information. Authors may choose some of the pictures to illustrate the differences among landscapes/freshwater systems.

l 100: Sentence structure

l 102: how many samples were collected each year and what would be there contribution to mean values? If there is temporal heterogeneity in sampling, mean values may differ with both spatial and temporal evolution.

From l 103: how many samples did you analyse for dissolved CO2 and CH4 and how

many did you use to calculate fluxes?

l 135: same title for 2.3 and 2.2 In general how are analysis and calculation representative?

From l 160: do differences in sampling frequencies have consequences to compare dissolved gas concentrations and fluxes to biogeochemical functioning? For example, for the ponds 10, 11, 16, chemical analyses were only performed on samples collected from 2010. In 5 years, pond conditions may have significantly changed with the important climate change in this region.

In Results

Figure 2: Not useful, could be put in supplementary data. Moreover, what represent the numbers between brackets? If they represent the number of samples, how authors can compare some sites with 15 samples and some sites with only 1? Noteworthy is the close relationship between Shoreline and Meltwater ponds, closer than Lake Hazen shoreline.

Table 2: How many samples for each pond type (not lake type)? Standard deviation should also be added. Authors my also provide mean and SD of the different physical and chemical parameters for each pond in order to compare with group values (in supplementary data for example). TDN could be added.

l 189: Illustrate the sentence 'without extremes during the growing season' with a figure;

l 195: Ammonia is not only produced in anoxic conditions, 'reduced ions' could be rephrased as ammonia or nutrient or inorganic ions, mineralization products...

l 196: How the Table S4 shows the increase in concentration of $NH_4$ with chemical change during the onset of flooding?

l 196: Both spatial and temporal aspects are used in cluster analysis. This may not be representative due to the discrepancy in sampling.

l 200-201: Inference from results that may be placed in discussion

Figure 3: The figure is not clear, seasonal trends are not clear, differences among sites and years are difficult to see. Scales of vertical axes could be modified according to maxima and minima values, especially for CH4 in meltwater ponds and lake Hazen shoreline. Lines between dots for Evaporative ponds should be removed; authors do not know what occur between their measurements. Evaporative ponds exhibit significantly less measurements than other sites. Results from this figure are not well explained and explored. Only cited twice at the beginning of 3.2.1 and 3.2.2, but not any arguments are based on this figure. Authors do not develop the seasonal trend of dissolved CO2 ad CH4. Comparison between years would be better highlighted using bars plots or a simple table.

Figure 4: Unclear, concentrations and fluxes should not be placed together in the same graph. Comparison between concentrations and fluxes in ponds are difficult. I advise to place dissolved gas concentrations in a graph and fluxes in another.

l 205-206: Although dissolved CO2 concentrations showed non-significant differences, authors compared these values between system types.

l 209: same comment as line 195.

l 221-222: CO2 and O2 correlation and relationship with water temperature not well showed in the Figure 5. Correlation coefficients may be placed in the main manuscript.

l 233-234: The sentence is not clear.

l 245-246: Still not any significant differences among pond types, but authors compared shoreline ponds values to other systems (l 255).

l 269-273: These are not results from this study, should be placed in discussion.

l 277: Is the assumption of generalization relevant and representative of the mean lake composition?

Figure 6: CO2 and CH4 fluxes should be separated. Vertical scales should be modified, for most ecosystems CO2 flux values cannot be read. Figure 6b may be change to a table. Although units were different, CO2 and CH4 fluxes have been already shown in Figure 4. This figure should be modified.

In the discussion

l 300: 'other compounds' is not clear

l 302: 'considerable' is a bit excessive considering dissolved CO2 concentrations

l 303: Are there evaporates in Lake Hazen watershed? Do you think weathering of carbonates is higher in Evaporative lake than in other systems (pH almost similar in all ponds)? Can DIC be released from surface water exhibiting pH around 8.3? This sentence is too general, higher CO2 concentration originates from higher microbial decomposition or as you write after due to concentration effects.

l 311: Do you have evidence of pond stratification other than correlation between CO2 and CH4 concentrations?

l 315: Associations may be replaced by correlations.

l 316-318: How do you evidence that productivity of microbial decomposition where not the main drivers? Both primary productivity and microbial activity could increase with temperature during the day and lead to diurnal O2 and CO2 concentration trends following temperature.

l 320-321: rephrase 'pre to post-flooding mean chl-a concentrations of 1.2 to 0.4 $\mu$g l-1)

l 324: 'reduce compounds' could be rephrase as nutrients or ammonia/nitrates, ammonia is not only produced in reduce conditions.

l 325-326: The sentence is not clear. Moreover, how diurnal O2 and CO2 concentration trends suggest that primary productivity was consistently occurring in Shoreline while

you seem to suggest the opposite l 316-318?

l 336-340: How can you evidence that $SO_4^{2-}$ production outcompeted $CH_4$ production? Maybe the locations of $SO_4^{2-}$ and $CH_4$ productions were different or the anoxia could not sustain methanogenic bacteria activity. Do you have measurements of dissolved $O_2$ or redox potential in the ponds?

l 344-345: The sentence is not clear, rephrase.

l 354-355: Are you sure (to your knowledge)?

Table 3 (l 357): Considering the intense Arctic change these last 25 years, how the compilation of data of $CO_2$ and $CH_4$ fluxes throughout more than twenty years can be relevant? Moreover, $CO_2$ and $CH_4$ fluxes may mostly differ according to soil nature, moisture, vegetation, microtopography or local climate conditions and not as a function of large latitudinal regions. I do not think this table provide useful and accurate information. Few words about the comparison between the measurements from this research and other studies would be enough. The main information provided by the table is also not clear.

Paragraph 4.3: This paragraph is too general; no specific point from your study is developed. Only few references are used to support your discussion (4 references, of which one is an article from authors). This entire paragraph should be modified: the discussion should be more based on your results, a specific and original point of view should be developed and your findings better compared with more articles.

---

## Referee Comment (RC2) · F. Joos (Referee) · 5 Jul 2016

Dear Authors

Thank you for submitting your work to Biogeosciences.

Please apologize the long processing time of your manuscript. Unfortunately, it was extremely difficult to find reviewers for your manuscript. I nominated in total 23 reviewers. Finally, two agreed to assess your manuscript. However, only one delivered a report while the other reviewer did neither respond to the automatically generated reminders nor to my personal e-mails asking for the referee report. Overall, these circumstances

lead to a lengthy review process and I am forced now to base my decision on a single review and my own reading. I regret that your colleagues appear not to be willing to review your work. Formally, I submit this comment now as a review to close the open discussion process.

The reviewer calls for major revision of your manuscript. On the positive site, the reviewer notes that your study shows interesting results about the weak $CO_2$ and $CH_4$ emissions and uptake from freshwater systems in High Arctic region and that such results are crucial to better constrain the assessment of future carbon feedback from permafrost environments. Among other comments, the referee calls for a better investigating of the biogeochemical processes responsible of contrasted $CO_2$ and $CH_4$ concentrations and fluxes and for a strongly improved discussion. The referee also notes a lack of references in certain sections and calls for improved figures and provides several suggestions how to improve the figures and the text. Please also check the numbering sections and subsections in the revised manuscript (section 4.2 seems missing).

I would like to encourage you to submit a detailed point by point response to each of the reviewer's comment with your answers pasted below the individual comments. Your response should detail how you address the criticism in the revised manuscript. I also ask you to prepare a revised manuscript where all your changes are highlighted in track change and to upload this track-changed manuscript as a pdf file as part of your reply.

The open discussion will now be closed after the posting of this editorial comment and I will proceed with a decision as soon as I will have received your reply to the review.

With kind regards,

Fortunat Joos

---

## Author Comment (AC1) · 2 Aug 2016

**bg-2016-79-RC1, 2016 Response to Reviewers Document**

Authors note (August 2016): We appreciate the close attention that Anonymous Referee #1 provided on their review of this paper. Their suggestions, we believe, substantially improve this manuscript, especially the Discussion section. We have incorporated nearly all of the suggested edits that the Reviewer provided and discuss why we propose not to incorporate others. We would like to take this opportunity to thank Reviewer #1. We have left Reviewer comments in regular type and our responses in ***bold italics***.

**Anonymous Referee #1**

General comments on the overall quality of the paper

In this study, authors examine CO2 and CH4 exchange between four common freshwater systems located in the high Arctic watershed of Lake Hazen, Elsmere Island, Nunavut, Canada. This study aims to measure net exchange of CO2 and CH4 between atmosphere and common high Arctic freshwater ecosystems. In previous studies, authors investigated CO2 and CH4 net exchanges from terrestrial ecosystems (e.g. polar semi-desert, meadow wetlands, uplands) in the watershed of Lake Hazen. In this study, authors also aim to contextualize their new findings about freshwater systems in Lake Hazen watershed with both their previous results and literature. The scientific context and questions are well defined but specific issues about High Arctic landscapes could be clarified. What are the conditions and issues in High Arctic environments that differ from Low Arctic and Sub Arctic regions? ***We have attempted to address this in a revised Introduction section (see below).***

The objectives are well exposed and correspond to data showed by authors. However, because of too general sentences and a lack of references, the answer of the second objective (contextualization) is not well developed and not accurate enough. ***This has been addressed in a new section 4.2 in the Discussion (see below).*** The authors present interesting data about CO2 and CH4 concentrations and exchange with atmosphere in freshwater systems in a high Arctic watershed. Measurements have been performed during summers from 2005 to 2012 in four freshwater ecosystems: evaporative ponds, meltwater ponds, shoreline ponds and Lake Hazen shoreline. These four freshwater systems were identified using a hierarchical clustering analysis based on gas concentrations and biogeochemistry data. The main findings from this study are: Mean CO2 concentrations and atmospheric exchanges were statistically similar between freshwater systems. The three types of ponds were weak CO2 emitters. Shoreline ponds exhibited the highest dissolved CH4 concentrations and fluxes to the atmosphere. However, because shoreline ponds cover a small area, their contribution to overall CH4 emissions was weak. Other ponds and the Lake Hazen shoreline showed similar CH4 concentrations and fluxes, which were weak. The same authors evidenced in previous studies that polar semi-desert ecosystems were weak sink of atmospheric CO2 but they significantly consumed CH4. Alternatively, meadow wetlands were important sink of CO2 and weak emitter of CH4. Considering their cover surface, all freshwater systems did not significantly contribute to total net C exchange from Lake Hazen watershed. This study showed interesting results about the weak CO2 and CH4 emissions and uptake from freshwater systems in High Arctic region. These results are crucial to better constrain the assessment of future carbon feedback from permafrost environments with climate change. Furthermore, high values of CH4 concentrations and fluxes

in shoreline ponds are important considering likely evolution of both the water level and the biogeochemistry of Arctic lakes. From my point of view, the main result is the seasonal flooding of Lake Hazen that led to strong increase in CO2 and CH4 emissions from ponds bordering the lake while other ecosystems were weak CH4 emitter. The highlighted processes are interesting and important although more evidences of the impact of biogeochemistry change on CH4 emissions would be necessary.

*Yes, we agree that the flooding story is interesting and may warrant further investigation with more targeted studies. However, an important goal of this work was to weight the relative importance of different ecosystems within watershed-scale exchange of carbon GHGs. We found that shoreline ponds were insignificant contributors to regional GHG exchange because of their low abundances in the watershed. We suggest in the new Discussion section 4.2 that future changes in the regional climate will likely not considerably affect the net exchange of carbon GHGs from Shoreline systems. Therefore, we have not focused on particular results from the Shoreline systems relative to other ponds and lakes.*

The second part of the discussion infers about the evolution of carbon exchange from freshwater systems in warmer and wetter conditions due to climate change. This section (paragraph 4.3) should be strongly modified. The current discussion section is too general and not adequately based on the findings from this study. There is a strong lack of references in this section (only four, an one auto-citation). Authors should develop a more specific and accurate discussion using more references. Despite the importance of data, authors are strongly recommended to do a major revision before acceptation by 1) better investigating the biogeochemical processes responsible of contrasted CO2 and CH4 concentrations and fluxes among ponds and 2) strongly improving the discussion. ***Both points have been addressed below.***

Scientific questions and issues

- I would recommend changing the title that does not reflect the main findings
*We have altered the manuscript title to more appropriately focus on the approach and findings of our study: "Carbon dioxide and methane fluxes of freshwater systems in the rapidly changing high Arctic".*

- The number of samples should be indicated. The standard deviation of the mean should be indicated.
*The number of samples issue has been addressed with the new Table 2, and revised Table 3 (formerly Table 2) which now includes means and standard deviations.*

How are representative the different measurements considering the differences in quantity of samples? How evolved the number of samples during time from 2005 to 2012? How many samples per site did you use to build the dendrogram?
*In the methods section "Numerical analysis" we discuss the advantages of using both hierarchical cluster analysis and linear mixed models for unbalanced sampling programs. We have now included a new Table 2 to show the number of samples taken for both greenhouse gases measurements and general chemistry, which were used in the cluster analysis.*

- Are the ponds permanent throughout the year? Do you consider these freshwater systems as ponds of small lakes?
*We have included the word "permanent" after "several" in the first sentence of the second paragraph of section 2.1. We have named our sites based on the type of water body they are (pond or lake).*

- What is the geologic substrate and the soil nature in the watershed? It may help to discuss your interesting results.
*We have modified the second sentence of section 2.1 to include after "(6,901 km$^2$),", "composed of carbonate, evaporite and dolomite rock (Trettin, 1994) and crysolic soils."*

- Place the section 'numerical analysis' currently located in the supplementary data in the main manuscript.
*We have now placed the "numerical analysis" section into the main manuscript at the end of the Methods section.*

- Authors studied both spatial and temporal variability. The two perspectives are not clearly exposed. I would recommend separating results about spatial variability from temporal evolution of gas concentrations and fluxes. The robustness of the spatial variability should be better explained by improving Figure 2 and Table 2. The available samples/data and the significance of differences in biogeochemical composition should be added. Some temporal trends should be better illustrated and explained (Figure 3).
*This has been addressed below.*

- This manuscript reproduces and repeats some results already presented in Emmerton et al. (2014). Results from previous studies should be removed from the abstract and from the result section.
*We have modified the results section (see below). We have also removed two sentences from the abstract which present terrestrial gas exchange data, and have added "and data from previous studies" after "When using ecosystem-cover classification mapping".*

- Most of the figures should be modified in order to clarify the main information. Concentrations and fluxes bar plots should be separated; vertical scales should be changed. Some figures in supplementary data could be placed in the main manuscript such as Figure S2.
*See responses below. Figure S2 is solid support for our lake classification, but we feel it is not a significant part of our results or discussion, especially considering that we moved Figure 2 to the SI.*

- Authors highlighted interesting biogeochemical processes, which could be better evidenced
*We have attempted to expand explanations of our CO2 and CH4 concentrations and fluxes using biogeochemical processes inferred by correlations in Table S4. However, we were cognizant of limiting these statements because of their speculative and correlative nature. We have added references and interpretations of CO2 and CH4 concentrations and fluxes within Results sections 3.2.1 and 3.2.2, and Discussion sections 4.1.1 and 4.1.2.*

- This manuscript requires a substantial improvement of the **section 4.3**. Scientific arguments should be more specific and based on the findings from this study.
*See below*.

Technical corrections

In the introduction:
From 35-36: The paragraph seems to be general and does not provide precise information about the weight of each process or where do they mainly occur? Is there a latitudinal gradient from Sub-Arctic to Low-Arctic and High-Arctic.
*We have attempted to increase the focus of the introduction on the differences between low Arctic and high Arctic greenhouse gases exchange. Most of the section has been re-written and re-organized with additions of new information.*

41-42: Check sentence structure
*This sentence has been modified to, "Due to its poor solubility in water, $CH_4$ can then be effectively released to the atmosphere from these ecosystems by ebullition and wind turbulence, perhaps contributing up to 12% of global emissions (Lai, 2009; Walter et al., 2006)."*

45-46 and 51: The freshwater systems cover more than 50% of area in northern regions but less than 5% in polar semi-desert landscapes? Authors may better explain this important difference between the general point of view and specific semi-desert landscapes, and could describe the latitudinal/landscape gradient?
*In the third sentence of the Introduction, we now specify that, "Northern latitudes, between approximately 45 and 75 °N, contain the highest abundance of lakes, ponds and wetlands on the planet (Lehner and Doll, 2004) due to historical glaciations and moderate annual precipitation." In the third paragraph of the Introduction, we now specify that, "In the high Arctic (>~70°N; AMAP), lake abundance and area are dramatically reduced on the landscape. The prevalence of cold and dry high pressure air masses results in a semi-arid climate with relatively well-drained and unproductive inorganic soils (Campbell and Claridge, 1992). This environment, therefore, discourages surface water retention with often less than 5% of the landscape being covered by aquatic systems" After each of these statements, we now provide more detailed information about the landscapes with respect to carbon GHG exchange.*

61-62: Control the relevance of chosen references, particularly Peterson et al. (2002) about permafrost thaw and Manabe et al., (1994) about growing seasons.
*These references have been replaced with Froese et al. 2008 (for Peterson et al.) and Myneni et al. 1997 (for Manabe 1994). Euskirchen 2007 replaced Froese et al. 2008.*

68: the reference (Antony et al., 2014) does not correspond to the sentence about polar semidesert.
*This sentence has been replaced, including the original reference.*

68-70: Sentence is not clear, check its structure

70: although it is uncertain how rapid climate change will alter the C cycle in northern landscapes, this study does not provide strong information about its evolution.

*These two sentences have been replaced with the following: "However, the net result of these processes on high-latitude freshwater carbon GHG exchange is not well delineated, nor is the relative contribution of freshwater systems to total landscape carbon GHG exchange. This information, from a rapidly changing and extensive biome (>10$^6$ km) is critical for improved global carbon models and budgeting."*

In Methods

Figure 1: the figure should be modified. The general maps are too small and thus not useful, the north arrow and the scale are also too small and not visible. Even on a half page, pond and lake pictures are small and don not provide any information. Authors may choose some of the pictures to illustrate the differences among landscapes/freshwater systems.

*Figure 1 has now been modified as per the reviewer's suggestions.*

100: Sentence structure

*This sentence has been modified to, "We also sampled shoreline water of Lake Hazen which potentially interacted with ponds located adjacent to its shoreline."*

102: how many samples were collected each year and what would be there contribution to mean values? If there is temporal heterogeneity in sampling, mean values may differ with both spatial and temporal evolution.

*We have now included a new Table 2 which is similar to Table S2, but shows only individual sample numbers of greenhouse gas collections, and full chemistry collections. We also bring more attention to the spatial imbalance of sampling. We include a reference to the new Table 2 by re-writing the last few sentences of section 2.1 to read, "Due to logistical issues related to accessing this remote area over consistent time periods each year, and due to the distances of some ponds from base camp, we completed an overall unbalanced sampling program in space and time. As a result, we focused on delineating biogeochemical differences between different types of high Arctic lakes, rather than on inter-annual biogeochemical trends within lakes. Regardless, all sampling occurred during the summer growing seasons of 2005 to 2012 (except for 2006), between mid-June and early August (Table 2, S2).*

From 103: how many samples did you analyse for dissolved CO2 and CH4 and how many did you use to calculate fluxes?

*We have now included a new Table 2 showing the frequency and year of CO2 and CH4 sampling, as well as chemistry sampling. The caption states that fluxes were calculated based on concentration sampling.*

135: same title for 2.3 and 2.2

*This has been fixed, 2.3 should have stated "fluxes" rather than "concentrations".*

In general how are analysis and calculation representative?

*There were two main analyses using results from multiple lake sites within different lake types. The hierarchical cluster analysis used each full biogeochemical sampling effort from each lake. Some lakes were sampled more intensively than others, so this would work as a conservative approach against finding defined groupings (e.g., there is potential for high biogeochemical variability between lakes sampled fewer times, so these should not separate from each other in the HCA in a consistent manner, unless differences were large). However, because the lake types represent such strongly different chemistries, our HCA results were well-defined, despite the higher chances of having hard to interpret results. We have added a sentence in the methods describing how our unbalanced approach should be conservative against finding well-separated groupings. The linear mixed model internally adjusts to unbalanced designs by using means of individual sites to find a representative group mean. Therefore this approach, as explained in the methods in Section 2.5, should be a solid technique for quantifying differences in CO2 and CH4 concentrations and fluxes between the lake types.*

From 160: do differences in sampling frequencies have consequences to compare dissolved gas concentrations and fluxes to biogeochemical functioning? For example, for the ponds 10, 11, 16, chemical analyses were only performed on samples collected from 2010. In 5 years, pond conditions may have significantly changed with the important climate change in this region.

*Because of logistical constraints which resulted in an unbalanced sampling design in time and space, we have focused on comparing lake types, rather than changes over time, or changes between individual lakes. The reviewer is correct that interannual changes in climate and possibly lake chemistry could have been considerable over a several year time scale, however by focusing on the large differences between lake types and their general time series patterns in our results and interpretations, we circumvent many of the difficulties associated with the unbalanced approach. Further, when comparing dissolved gases and chemistry between lake types, we only used concurrent samples when gases and chemistry were taken together.*

In Results

Figure 2: Not useful, could be put in supplementary data. Moreover, what represent the numbers between brackets? If they represent the number of samples, how authors can compare some sites with 15 samples and some sites with only 1? Noteworthy is the close relationship between Shoreline and Meltwater ponds, closer than Lake Hazen shoreline.

*This figure has been moved to SI. We have updated the caption to explain the numbers in the brackets, which are the number of nodes (samples) compressed by site for ease of display. Please see responses above and below about comparing samples of different sizes. We agree that the closer relationship between Shoreline and Meltwater ponds is interesting, and likely driven by the fact they are each small systems. However, we are having difficulties finding a suitable place within the paper to discuss these more resolved results and setting them within the much larger context of watershed-scale gas exchange.*

Table 2: How many samples for each pond type (not lake type)? Standard deviation should also be added. Authors my also provide mean and SD of the different physical and chemical parameters for each pond in order to compare with group values (in supplementary data for example). TDN could be added.
*Please see the new Table 2 for number of samples taken for each water body. Standard deviations have been added to the former Table 2 (now Table 3). Table 3 now has mean and SD for each pond and TDN has been added to the table.*

189: Illustrate the sentence 'without extremes during the growing season' with a figure;
*The statement, "without extremes during the growing season" has been supplemented with, "(see section 3.2)." at the end of the sentence. This section introduces the time series figure.*

195: Ammonia is not only produced in anoxic conditions, 'reduced ions' could be rephrased as ammonia or nutrient or inorganic ions, mineralization products: : :
*This sentence has been removed from the manuscript.*

196: How the Table S4 shows the increase in concentration of NH4 with chemical change during the onset of flooding?
*We have now added pre- and post-flooding chemistry into the former Table 2 (now Table 3). We have updated the reference to Table S4 to Table 3. We now focus on $NO_3^-+NO_2^-$ as an indicator of chemical change during flooding.*

196: Both spatial and temporal aspects are used in cluster analysis. This may not be representative due to the discrepancy in sampling.
*By using the cluster analysis with multiple sites over multiple years, we are maximizing variation within and between sites, and increasing the potential for spurious organization of the freshwater systems. However, based on the results and the chemical differences between groups of lakes (new Table 3), it was evident that between-lake chemistry differences outweighed within-lake differences between years. We have modified the "Numerical analysis" section by adding after the first sentence, "Because sampling was unbalanced in frequency and time between sites due to logistical challenges (Table 2; see section 2.1), potential overlap of chemistries between individual lakes was high, therefore setting a conservative standard for classifying distinct lake types."*

200-201: Inference from results that may be placed in discussion
*This result or interpretation is not central to this paper, so the sentence has been removed.*

Figure 3: The figure is not clear, seasonal trends are not clear, differences among sites and years are difficult to see. Scales of vertical axes could be modified according to maxima and minima values, especially for CH4 in meltwater ponds and lake Hazen shoreline. Lines between dots for Evaporative ponds should be removed; authors do not know what occur between their measurements. Evaporative ponds exhibit significantly less measurements than other sites.
*At the outset of this study, our aim was to group and describe different types of high Arctic aquatic systems. Each year, the timing and extent of our visits changed due to logistical difficulties and the timing of other studies on site, so interannual differences within systems were difficult to delineate. We felt, therefore, that by standardizing the axes of the graph, we*

*would better highlight differences between the lake types. We believe this also allows the reader to more easily see the general intra-annual trend for each lake type. For CH4, this meant that only the Shoreline ponds showed important trends, which was much of our point in our discussion anyways. We have removed lines between the dots in all graphs. Unbalanced sampling design has been discussed in other responses above.*

Results from this figure are not well explained and explored. Only cited twice at the beginning of 3.2.1 and 3.2.2, but not any arguments are based on this figure. Authors do not develop the seasonal trend of dissolved CO2 ad CH4. Comparison between years would be better highlighted using bars plots or a simple table.
*We realize that the beginning sentence of section 3.2.1, where Figures 3 and 4 are referenced, may de-emphasize the role the figures play in presenting the results through the balance of the section. We have strategically placed more references to Figures 3 and 4 throughout the balance of section 3.2.1 so as to clarify the importance of these figures. We do structure arguments later on based on the general trends in GHG exchange and differences between systems shown in Figure 3. However our sampling logistics would not allow for meaningful inter-annual comparisons within lake types, though that was not the aim of this study. We would prefer to keep Figure 3 as is, because bar plots, tables or individually-scaled axes would over-emphasize interannual differences, rather than between-type differences which were the focus of this study. However, we have added this same figure, fully-scaled for each site, in the SI (Figure S4). We have also removed connecting lines in the plots.*

Figure 4: Unclear, concentrations and fluxes should not be placed together in the same graph. Comparison between concentrations and fluxes in ponds are difficult. I advise to place dissolved gas concentrations in a graph and fluxes in another.
*This figure has been modified so that the upper panel shows only concentrations of CO2 and CH4, while the lower panel shows only fluxes of CO2 and CH4.*

205-206: Although dissolved CO2 concentrations showed non-significant differences, authors compared these values between system types.
*We have stated at the beginning of the section that the differences were not statistically significant. However, we feel it is still a useful exercise to compare the systems because, for example, the relatively fewer samples from the Evaporative ponds could have possibly influenced their high variability and therefore non-significant differences with the other systems, which were sampled more consistently.*

209: same comment as line 195.
*This sentence has been modified to, "These ponds were the shallowest of the four sampled and were rich in dissolved iron, DIC, and , TDP."*

221-222: CO2 and O2 correlation and relationship with water temperature not well showed in the Figure 5. Correlation coefficients may be placed in the main manuscript.
*We have updated this sentence to reflect the weak relationship between CO2 and O2, but strong association between CO2 and water temperature, "Mean diurnal trends in $CO_2$ concentrations across all sampling years, as measured by the automated system at Skeleton Lake, showed that $CO_2$ and $O_2$ concentrations had little association together (Pearson*

*correlation: r= -0.18, df=7, p=0.67), but CO₂ associated strongly and negatively with water temperature (r=-0.97, df=7, p<0.001; Figure 5)." We have also updated section 4.1.1, to reflect the poor association between CO2 and O2, "Further, mean diurnal CO₂ and O₂ concentrations in surface waters associated poorly together, rather than oppositely if metabolic processes (i.e., primary productivity or decomposition of organic matter; see Pond 01 below) were dominant drivers in surface waters." We have updated the same results for Pond1 in the following results paragraph to, "Diurnal trends of CO₂ and O₂ concentration measured by the automated system at Pond 01 over several growing seasons showed opposite diel patterns of the gases, with greater O₂ during the warmest and lightest parts of the day (r=-0.98, df=7, p<0.001; Figure 4)."*

233-234: The sentence is not clear. Clarify in text
*We have decided that this sentence should be removed for clarity purposes.*

245-246: Still not any significant differences among pond types, but authors compared shoreline ponds values to other systems (255).
*There are statistical differences between Shoreline Ponds (discussed in paragraph following this one) and the others with respect to CH4 concentrations (Figure 4). Therefore, it is prudent to discuss why Shoreline Ponds were different from the other systems.*

269-273: These are not results from this study, should be placed in discussion.
*The first sentence of this group has been placed in section 2.5 of the Methods section. The second sentence has been removed from the manuscript.*

277: Is the assumption of generalization relevant and representative of the mean lake composition?
*There are two reasons that this assumption may be valid. First, only a thin moat along the shoreline of Lake Hazen is exposed during many years if summer temperatures remain cool and wind storms are infrequent. During these years, lake-scale gas exchange would only occur in the shoreline areas. Second, more targeted work is ongoing at Lake Hazen that has uncovered evidence that shoreline gas exchange is comparable to pelagic regions of this ultra-oligotrophic lake. We have modified this sentence to include reference to unpublished data from 2015, "When assuming its shoreline waters were representative of the entire lake area as recent evidence suggests (unpublished data, 2015), the expansive Lake Hazen…".*

Figure 6: CO2 and CH4 fluxes should be separated. Vertical scales should be modified, for most ecosystems CO2 flux values cannot be read. Figure 6b may be change to a table. Although units were different, CO2 and CH4 fluxes have been already shown in Figure 4. This figure should be modified.
*This figure has now been replaced with a table of values and indications of statistical significance between ecosystem types.*

In the discussion

300: 'other compounds' is not clear

*We have updated this sentence to read, "Concentrations of $CO_2$ and other water chemistry measurements were highest in small…".*

302: 'considerable' is a bit excessive considering dissolved CO2 concentrations
*We have removed the word "considerable".*

303: Are there evaporates in Lake Hazen watershed? Do you think weathering of carbonates is higher in Evaporative lake than in other systems (pH almost similar in all ponds)? Can DIC be released from surface water exhibiting pH around 8.3? This sentence is too general, higher CO2 concentration originates from higher microbial decomposition or as you write after due to concentration effects.
*We have included in the methods (Section 2.1) a sentence that states evaporitic and carbonitic geology is prevalent in the watershed. Marce et al. 2015 (Nature Geoscience v8) demonstrate the potentially large contribution to CO2 supersaturation in lakes by carbonate in high alkalinity environments, such as those at our sites, though its magnitude is affected by water temperature. Evaporation-concentration of shallow evaporative lakes have helped push their alkalinities above 2 mEq L-1, unlike most other systems we studied, which were less than 2 mEq L-1 (except Pond 1: 2.3 mEq L-1). This may have amplified the contribution of carbonates to CO2 supersaturation in these lakes, but yes, overall we may not expect substantial weathering differences between lake types. We have added to this sentence to strengthen the argument that carbonate weathering played a role in CO2 concentrations in our lakes: "Dissolved $CO_2$ was likely being produced effectively in all Evaporative ponds by ecosystem metabolism because of their high concentrations of DOC. These, another other, isolated systems concentrate many solutes in their waters including degraded allochthonous and fresh autochthonous DOC (Tank et al., 2009), which would be available as a source of energy to heterotrophs. Accumulation and dissociation of weathered carbonates and evaporates in these moderately warm, high alkalinity environments (2-5 mEq $L^{-1}$) may have also been important (Trettin, 1994; Marcé et al., 2015)."*

311: Do you have evidence of pond stratification other than correlation between CO2 and CH4 concentrations?
*Yes, in the previous sentence we refer to Figure S6, which shows stratification of Skeleton Lake, a Meltwater system.*

315: Associations may be replaced by correlations.
*"Associations" has been replaced by "correlations".*

316-318: How do you evidence that productivity of microbial decomposition where not the main drivers? Both primary productivity and microbial activity could increase with temperature during the day and lead to diurnal O2 and CO2 concentration trends following temperature.
*This is a good point and not one that our data could definitively solve. Our statement, rather, is supported by the strong, opposite diurnal patterns of CO2 and O2 observed in the visibly more productive Pond 01 compared to Skeleton Lake. Pond 01 supported a widespread emergent vegetation community compared to Skeleton Lake's benthic mat communities in the near-shore area. We have attempted to downplay our statement by modifying the sentence in question to, "Results from our automated systems supported this argument as mean diurnal*

*$CO_2$ and $O_2$ concentrations in surface waters of Skeleton Lake associated poorly together, rather than oppositely if metabolic processes (i.e., primary productivity or decomposition of organic matter; see Pond 01 below) were dominant drivers in surface waters."*

320-321: rephrase 'pre to post-flooding mean chl-a concentrations of 1.2 to 0.4 _g l-1)
*The bracketed item has been rephrased to, "(pre-flooding: 1.2 µg $L^{-1}$ chl-a; post-flooding: 0.4 µg $L^{-1}$ chl-a)"*

324: 'reduce compounds' could be rephrase as nutrients or ammonia/nitrates, ammonia is not only produced in reduce conditions.
*The second half of this sentence has been removed from the manuscript.*

325-326: The sentence is not clear. Moreover, how diurnal O2 and CO2 concentration trends suggest that primary productivity was consistently occurring in Shoreline while you seem to suggest the opposite l 316-318?
*Discussion of diurnal gas concentrations on lines 316-318 refers to trends observed in Skeleton Lake only. We have modified the unclear sentence to, "Although negatively correlated diurnal $CO_2$ and $O_2$ concentrations suggest that primary productivity was consistently occurring in Shoreline pond surface waters, seasonal flooding of the ponds was ultimately the more important process controlling seasonal $CO_2$ concentrations."*

336-340: How can you evidence that SO42- production outcompeted CH4 production? Maybe the locations of SO42- and CH4 productions were different or the anoxia could not sustain methanogenic bacteria activity. Do you have measurements of dissolved O2 or redox potential in the ponds?
*This statement is indeed somewhat speculative based on the lack of site-specific data from the sediments. We use anecdotal evidence to suggest that sulfate-reduction was the important anoxic process occurring in the sediments based on low CH4 concentrations in the lake's waters, a strong H2S smell from sediments in extracted cores (results unpublished to date), and the relative lack of ebullition fluxes from the lake. We would expect at least some evidence of higher CH4 fluxes during or after wind events over the extensive sampling record of Skeleton if location of methanogenic CH4 production was simply distributed in space according to micro-conditions in the sediments. Our water boxes did measure dissolved O2 in surface waters, which were relatively high, but unfortunately we do not have such measurements from the sediments. Since we qualify our statement in the proceeding sentence, we believe that the statement, though somewhat speculative, still provides some value to the interpretation of the results by at least highlighting the high concentrations of SO4 in these upland systems. The new sentences read, "Evaporative and Meltwater systems were typically weak producers and emitters of $CH_4$, which was possibly related to concurrently high $SO_4^{2-}$ concentrations in these systems due to additions of water draining evaporite geologies (Table 3; Trettin, 1994). This may have given competitive advantage to $SO_4^{2-}$-reducing bacterial communities in sediments, which typically outcompete methanogenic bacteria for hydrogen."*

344-345: The sentence is not clear, rephrase.
*This sentence has been modified to, "Only during periods of strong wind mixing of surface waters, or when Shoreline ponds breached and mixed organic particles (Table S4) across its*

*shoreline, did the near shore waters of Lake Hazen release CH₄ to the atmosphere above near-zero values."*

354-355: Are you sure (to your knowledge)?
*We have reviewed several studies of terrestrial and aquatic systems, especially from the high Arctic, and most sites and programs do not investigate fluxes from aquatic and terrestrial systems concurrently. However, we will put the caveat "To our knowledge," at the start of the third sentence in Section 4.2.*

Table 3 (l 357): Considering the intense Arctic change these last 25 years, how the compilation of data of CO2 and CH4 fluxes throughout more than twenty years can be relevant? Moreover, CO2 and CH4 fluxes may mostly differ according to soil nature, moisture, vegetation, microtopography or local climate conditions and not as a function of large latitudinal regions. I do not think this table provide useful and accurate information. Few words about the comparison between the measurements from this research and other studies would be enough. The main information provided by the table is also not clear.
*We agree with the reviewer that Table 3 should be removed, and this has been done. We have added elements of our findings from this table into other portions of the Discussion, including in the first paragraph of section 4.2.*

Paragraph 4.3: This paragraph is too general; no specific point from your study is developed. Only few references are used to support your discussion (4 references, of which one is an article from authors). This entire paragraph should be modified: the discussion should be more based on your results, a specific and original point of view should be developed and your findings better compared with more articles.
*Section 4.2 (formerly incorrectly numbered 4.3) has been completely re-written with improved focus on our results, with more specific original discussion points and with more complete referencing.*
* * *
**Reviewer #2**

…better investigating of the biogeochemical processes responsible of contrasted CO2 and CH4 concentrations and fluxes and for a strongly improved discussion.
*We have attempted to improve some of the biogeochemical interpretations, as outlined above. Discussion section 4.2 has been re-written and improved (see reviewer response above).*

…Lack of references in certain sections and calls for improved figures and provides several suggestions how to improve the figures and the text.
*Referencing in the last discussion section (4.2) has been improved with the re-write. Figures have been improved based on Reviewer #1 comments. We prefer not to make the suggested changes in the old Figure 3 (now Figure 2), please see reviewer response above. Text-based suggestions by Reviewer #1 have been made (see above).*

Please also check the numbering sections and subsections in the revised manuscript (section 4.2 seems missing).
*This has been fixed appropriately.*